# Odd-paired controls frequency doubling in *Drosophila* segmentation by altering the pair-rule gene regulatory network

**Erik Clark\*, Michael Akam**

Laboratory for Development and Evolution, Department of Zoology, University of Cambridge, Cambridge, United Kingdom

**Abstract** The *Drosophila* embryo transiently exhibits a double-segment periodicity, defined by the expression of seven 'pair-rule' genes, each in a pattern of seven stripes. At gastrulation, interactions between the pair-rule genes lead to frequency doubling and the patterning of 14 parasegment boundaries. In contrast to earlier stages of *Drosophila* anteroposterior patterning, this transition is not well understood. By carefully analysing the spatiotemporal dynamics of pair-rule gene expression, we demonstrate that frequency-doubling is precipitated by multiple coordinated changes to the network of regulatory interactions between the pair-rule genes. We identify the broadly expressed but temporally patterned transcription factor, Odd-paired (Opa/Zic), as the cause of these changes, and show that the patterning of the even-numbered parasegment boundaries relies on Opa-dependent regulatory interactions. Our findings indicate that the pair-rule gene regulatory network has a temporally modulated topology, permitting the pair-rule genes to play stage-specific patterning roles.

## Introduction

Segmentation is a developmental process that subdivides an animal body axis into similar, repeating units (*Hannibal and Patel, 2013*). Segmentation of the main body axis underlies the body plans of arthropods, annelids and vertebrates (*Telford et al., 2008*; *Balavoine, 2014*; *Graham et al., 2014*). In arthropods, segmentation first involves setting up polarised boundaries early in development to define 'parasegments' (*Martinez-Arias and Lawrence, 1985*). Parasegment boundaries are maintained by an elaborate and strongly-conserved signalling network of 'segment-polarity' genes (*Ingham, 1988*; *Perrimon, 1994*; *DiNardo et al., 1994*; *Sanson, 2001*; *Janssen and Budd, 2013*).

In all arthropods yet studied, the segmental stripes of segment-polarity genes are initially patterned by a group of transcription factors known as the 'pair-rule' genes (*Green and Akam, 2013*; *Peel et al., 2005*; *Damen et al., 2005*). The pair-rule genes were originally identified in a screen for mutations affecting the segmental pattern of the *Drosophila melanogaster* larval cuticle (*Nüsslein-Volhard and Wieschaus, 1980*). They appeared to be required for the patterning of alternate segment boundaries (hence 'pair-rule') and were subsequently found to be expressed in stripes of double-segment periodicity (*Hafen et al., 1984*; *Akam, 1987*).

Early models of *Drosophila* segmentation suggested that the blastoderm might be progressively patterned into finer-scale units by some reaction-diffusion mechanism that exhibited iterative frequency-doubling (reviewed in *Jaeger, 2009*). The discovery of a double-segment unit of organisation seemed to support these ideas, and pair-rule patterning was therefore thought to be an adaptation to the syncytial environment of the early *Drosophila* embryo, which allows diffusion of gene products between neighbouring nuclei. However, the transcripts of pair-rule genes are apically localised during cellularisation of the blastoderm, and thus pair-rule patterning occurs in an

**\*For correspondence:** ec491@cam.ac.uk

**Competing interests:** The authors declare that no competing interests exist.

**eLife digest** The basic body plan of an animal is set up in the early embryo, where key developmental genes are expressed in specific patterns across the organism. These patterns emerge from the way in which the proteins encoded by these genes act to regulate each other's expression. The fruit fly *Drosophila* is often used as a simple model for studying how regulatory interactions between genes lead to the formation of complex developmental patterns.

One example is segmentation, the process by which the trunk (main body) region of the *Drosophila* embryo is subdivided into 14 segments. A group of transcription factor proteins that are encoded by the so-called "pair-rule" genes play a crucial role in producing the final pattern. Early in development, the pair-rule genes are expressed in patterns of seven stripes, dividing the embryo into double segment units. As the embryo develops, these patterns change to form more precise patterns of 14 stripes, corresponding to single segments. Although many of the regulatory interactions between the pair-rule genes were known, how the system works as a whole to produce this change in expression patterns was not well understood.

Clark and Akam have now examined how the patterns of pair-rule gene expression change over time inside fly embryos. This revealed that a "rewiring" of the network of pair-rule genes occurs at the end of the seven stripe stage to produce fourteen stripes. Further investigation revealed that a transcription factor encoded by the gene *odd-paired* causes this rewiring, and that the timing of the expression of the Odd-paired protein determines when the rewiring happens.

Further studies could now investigate whether Odd-paired's role as a timer extends to species where segments emerge sequentially, instead of the simultaneous formation of segments seen in *Drosophila*. Future challenges will be to find out how Odd-paired interacts with other pair-rule transcription factors, and whether there are other timing factors that help to coordinate embryonic patterning.

effectively cellular environment (*Edgar et al., 1987*; *Davis and Ish-Horowicz, 1991*). Furthermore, double-segment periodicity of pair-rule gene expression is also found in some sequentially segmenting ('short germ') insects (*Patel et al., 1994*), indicating that pair-rule patterning predates the evolution of simultaneous ('long germ') segmentation (*Figure 1*).

The next set of models for pair-rule patterning were motivated by genetic dissection of the early regulation of the segment-polarity gene *engrailed (en)*. It was found that odd-numbered *en* stripes – and thus the anterior boundaries of odd-numbered parasegments (hereafter 'odd-numbered parasegment boundaries') – require the pair-rule gene *paired (prd)*, but not another pair-rule gene *fushi tarazu (ftz)*, while the opposite was true for the even-numbered *en* stripes and their associated ('even-numbered') parasegment boundaries (*DiNardo and O'Farrell, 1987*). Differential patterning of alternate segment-polarity stripes, combined with the observation that the different pair-rule genes are expressed with different relative phasings along the anterior-posterior (AP) axis, led to models where static, partially overlapping domains of pair-rule gene expression form a combinatorial regulatory code that patterns the blastoderm with single-cell resolution (*DiNardo and O'Farrell, 1987*; *Gergen and Butler, 1988*; *Weir et al., 1988*; *Coulter et al., 1990*; *Morrissey et al., 1991*).

However, pair-rule gene expression domains are not static. One reason for this is that their upstream regulators, the gap genes, are themselves dynamically expressed, exhibiting expression domains that shift anteriorly over time (*Jaeger et al., 2004*; *El-Sherif and Levine, 2016*). Another major reason is that, in addition to directing the initial expression of the segment-polarity genes, pair-rule genes also cross-regulate one another. Pair-rule proteins and transcripts turn over extremely rapidly (*Edgar et al., 1986*; *Nasiadka and Krause, 1999*), and therefore regulatory feedback between the different pair-rule genes mediates dynamic pattern changes throughout the period that they are expressed. Most strikingly, many of the pair-rule genes undergo a transition from double-segment periodicity to single-segment periodicity at the end of cellularisation. The significance of this frequency-doubling is not totally clear. In some cases, the late, segmental stripes are crucial for proper segmentation (*Cadigan et al., 1994b*), but in others they appear to be

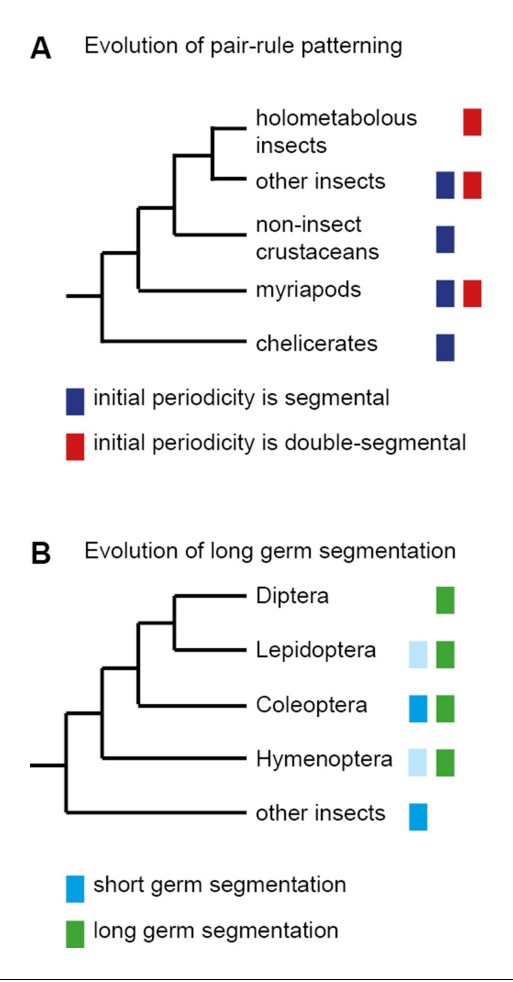

**Figure 1.** The evolution of pair-rule patterning predates the evolution of long germ segmentation. (**A**) Single segment periodicity is ancestral in arthropod segmentation, being found in spiders, millipedes, crustaceans and some insects (***Davis et al., 2005***; ***Pueyo et al., 2008***). 'Pair-rule' patterning, involving an initial double segment periodicity of pair-rule gene expression, appears to have evolved independently at least twice. It is found in insects and certain centipedes (***Davis et al., 2001***; ***Chipman et al., 2004***). (**B**) Long germ segmentation is likely to have evolved independently multiple times within holometabolous insects, from an ancestral short germ state (***Liu and Kaufman, 2005***). Light blue boxes for the Lepidoptera and Hymenoptera indicate that short germ segmentation is relatively uncommon in these clades.

dispensable (***Coulter et al., 1990***; ***Fujioka et al., 1995***), or their function (if any) is not known (***Klingler and Gergen, 1993***; ***Jaynes and Fujioka, 2004***).

More recent models of pair-rule patterning recognise that the pair-rule genes form a complex gene regulatory network that mediates dynamic patterns of expression (***Edgar et al., 1989***; ***Sánchez and Thieffry, 2003***; ***Jaynes and Fujioka, 2004***). However, whereas other stages of *Drosophila* segmentation have been extensively studied from a dynamical systems perspective (reviewed in ***Jaeger, 2009***; ***Grimm et al., 2010***; ***Jaeger, 2011***), we do not yet have a good systems-level understanding of the pair-rule gene network (***Jaeger, 2009***). This appears to be a missed opportunity: not only do the pair-rule genes exhibit fascinating transcriptional regulation, but their interactions are potentially very informative for comparative studies with short germ arthropods. These include the beetle *Tribolium castaneum*, in which the pair-rule genes form a segmentation oscillator (***Sarrazin et al., 2012***; ***Choe et al., 2006***).

To better understand exactly how pair-rule patterning works in *Drosophila*, we carried out a careful analysis of pair-rule gene regulation during cellularisation and gastrulation, drawing on both the genetic literature and a newly generated dataset of double-fluorescent in situs. Surprisingly, we found that the majority of regulatory interactions between pair-rule genes are not constant, but undergo dramatic changes just before the onset of gastrulation. These regulatory changes mediate the frequency-doubling phenomena observed in the embryo at this time.

We then realised that all the regulatory interactions specific to the late pair-rule gene regulatory network seem to require the non-canonical pair-rule gene *odd-paired (opa)*. *opa* was identified through the original *Drosophila* segmentation screen as being required for the patterning of the even-numbered parasegment boundaries (***Jürgens et al., 1984***). However, rather than being expressed periodically like the rest of the pair-rule genes, *opa* is expressed ubiquitously throughout the trunk region (***Benedyk et al., 1994***). The reported appearance of Opa protein temporally correlates with the time we see regulatory changes in the embryo, indicating that it may be directly responsible for these changes. We propose that Opa provides a source of temporal information that acts combinatorially with the spatial information provided by the periodically expressed pair-rule genes. Pair-rule patterning thus appears to be a two-stage process that relies on the interplay of spatial and temporal signals to permit a common set of patterning genes to carry out stage-specific regulatory functions.

# Results

## High-resolution spatiotemporal characterisation of wild-type pair-rule gene expression

We carried out double fluorescent in situ hybridisation on fixed wild-type *Drosophila* embryos for all pairwise combinations of the pair-rule genes *hairy, even-skipped (eve), runt, fushi tarazu (ftz), odd-skipped (odd), paired (prd)* and *sloppy-paired (slp)*. Because the expression patterns of these genes develop dynamically but exhibit little embryo-to-embryo variability (*Surkova et al., 2008*; *Little et al., 2013*; *Dubuis et al., 2013*), we were able to order images of individual embryos by inferred developmental age. This allowed us to produce pseudo time-series that illustrate how pair-rule gene expression patterns change relative to one another during early development (*Figure 2*).

The expression profile of each individual pair-rule gene has been carefully described previously (*Hafen et al., 1984*; *Ingham and Pinchin, 1985*; *Macdonald et al., 1986*; *Kilchherr et al., 1986*; *Gergen and Butler, 1988*; *Coulter et al., 1990*; *Grossniklaus et al., 1992*), and high-quality relative expression data are available for pair-rule proteins (*Pisarev et al., 2009*). In addition, expression atlases facilitate the comparison of staged, averaged expression profiles of many different blastoderm patterning genes at once (*Fowlkes et al., 2008*). However, because the pair-rule genes are expressed extremely dynamically and in very precise patterns, useful extra information can be gleaned by directly examining relative expression patterns in individual embryos. In particular, we have found these data invaluable for understanding exactly how stripe phasings change over time, and for interrogating regulatory hypotheses. In addition, we have characterised pair-rule gene expression up until early germband extension, whereas blastoderm expression atlases stop at the end of cellularisation.

Our entire wild-type dataset (32 gene combinations, >600 individual embryos) is available from the Dryad Digital Repository (*Clark and Akam, 2016*). We hope it proves useful to the *Drosophila* community.

## Three main phases of pair-rule gene expression

We classify the striped expression of the pair-rule genes into three temporal phases (*Figure 3A*). Phase 1 (equivalent to phase 1 of *Schroeder et al., 2011*; timepoint 1 in *Figure 2*) corresponds to early cellularisation, before the blastoderm nuclei elongate. Phase 2 (spanning phases 2 and 3 of *Schroeder et al., 2011*; timepoints 2–4 in *Figure 2*) corresponds to mid cellularisation, during which the plasma membrane progressively invaginates between the elongated nuclei. Phase 3 (starting at phase 4 of *Schroeder et al., 2011* but continuing beyond it; timepoints 5–6 in *Figure 2*) corresponds to late cellularisation and gastrulation. Our classification is a functional one, based on the times at which different classes of pair-rule gene regulatory elements (*Figure 3B*) have been found to be active in the embryo.

During phase 1, expression of specific stripes is established through compact enhancer elements mediating gap gene inputs (*Howard et al., 1988*; *Goto et al., 1989*; *Harding et al., 1989*; *Pankratz and Jäckle, 1990*). *hairy, eve* and *runt* all possess a full set of these 'stripe-specific' elements, together driving expression in all seven stripes, while *ftz* lacks an element for stripe 4, and *odd* lacks elements for stripes 2, 4 and 7 (*Schroeder et al., 2011*). These five genes are together classified as the 'primary' pair-rule genes, because in all cases the majority of their initial stripe pattern is established de novo by non-periodic regulatory inputs. The regulation of various stripe-specific elements by gap proteins has been studied extensively (for example *Small et al., 1992*, *1996*).

Phase 2 is dominated by the expression of so-called 'zebra' (or '7-stripe') elements (*Hiromi et al., 1985*; *Dearolf et al., 1989*; *Butler et al., 1992*). These elements, which tend to be relatively large (*Gutjahr et al., 1994*; *Klingler et al., 1996*; *Schroeder et al., 2011*), are regulated by pair-rule gene inputs and thus produce periodic output patterns. The stripes produced from these elements overlap with the stripes generated by stripe-specific elements, and often the two sets of stripes appear to be at least partially redundant. For example, *ftz* and *odd* lack a full complement of stripe-specific elements (see above), while the stripe-specific elements of *runt* are dispensable for segmentation (*Butler et al., 1992*). Neither *hairy* nor *eve* appears to possess a zebra element, and thus their expression during phase 2 is driven entirely by their stripe-specific elements. (Note that the 'late' (or 'autoregulatory') element of *eve* (*Goto et al., 1989*; *Harding et al., 1989*) does generate a periodic

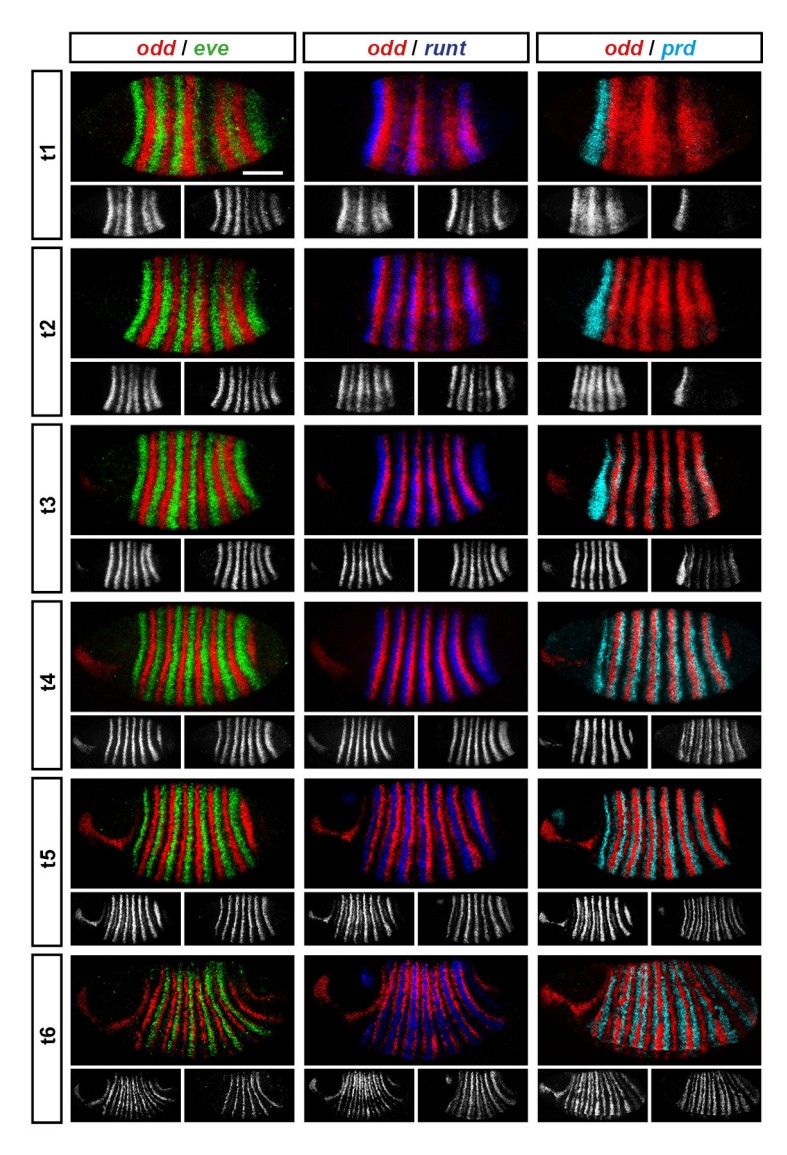

**Figure 2.** Representative double fluorescent in situ hybridisation data for three combinations of pair-rule genes. This figure shows a small subset of our wild-type dataset. Each column represents a different pairwise combination of in situ probes, while each row shows similarly-staged embryos of increasing developmental age. All panels show a lateral view, anterior left, dorsal top. Individual channels are shown in grayscale below each double-channel image. For ease of comparison, the signal from each gene is shown in a different colour in the double-channel images. Time classes are arbitrary, meant only to illustrate the progressive stages of pattern maturation between early cellularisation (t1) and late gastrulation (t6). Note that the developing pattern of *odd* expression in the head provides a distinctive and reliable indicator of embryo age. Scale bar = 100 μm. The complete dataset is available from the Dryad Digital Repository (*Clark and Akam, 2016*).

pattern and has therefore been considered to be analogous to the zebra elements of other pair-rule genes. However, because it is not expressed until phase 3 (*Schroeder et al., 2011*), we do not classify it as such.)

In addition to the five primary pair-rule genes, there are two other pair-rule genes, *prd* and *slp*, that turn on after regular periodic patterns of the other genes have been established. These genes possess only a single, anterior stripe-specific element, and their trunk stripes are generated by a zebra element alone (*Schroeder et al., 2011*). Because (ignoring the head stripes) these genes are

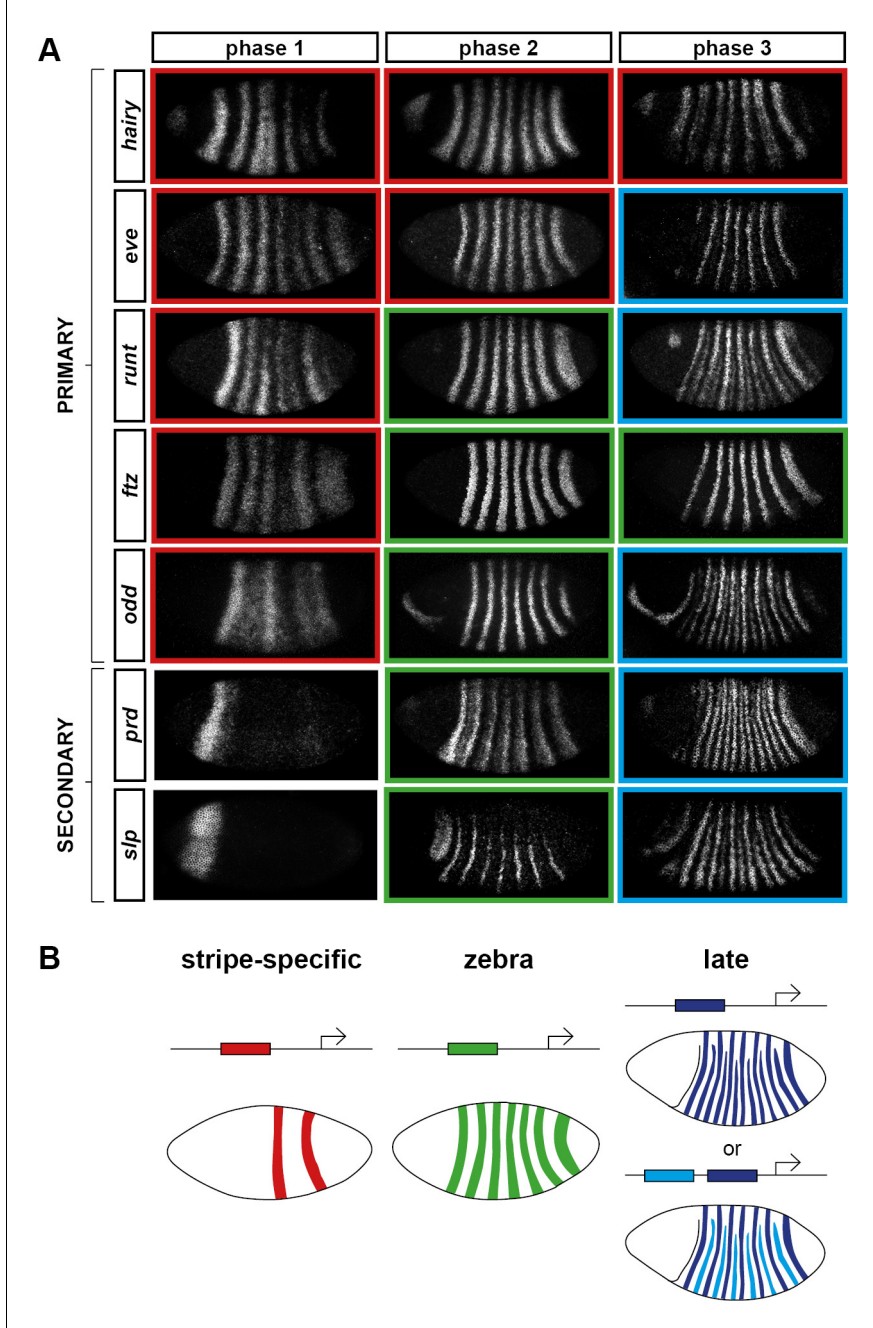

**Figure 3.** Three phases of pair-rule gene expression, usually mediated by different classes of regulatory element. (A) Representative expression patterns of each of the seven pair-rule genes at phase 1 (early cellularisation), phase 2 (mid cellularisation) and phase 3 (gastrulation). Pair-rule genes are classified as 'primary' or 'secondary' based on their regulation and expression during phase 1 (see text). All panels show a lateral view, anterior left, dorsal top. Note that the cephalic furrow may obscure certain anterior stripes during phase 3. (B) Illustrative diagrams of the different kinds of regulatory elements mediating pair-rule gene expression. 'Stripe-specific' elements are regulated by gap genes and give rise to either one or two stripes each. 'Zebra' elements are regulated by pair-rule genes and give rise to seven stripes. 'Late' expression patterns may be generated by a single-element generating segmental stripes, or by a combination of two elements each generating a distinct pair-rule pattern. The coloured outlines around the panels in (A) correspond to the colours of the different classes of regulatory elements in (B), and indicate how each phase of expression of a given pair-rule gene is thought to be regulated. See text for details.

regulated only by other pair-rule genes, and not by gap genes, they are termed as the 'secondary' pair-rule genes.

The third, 'late' phase of expression is the least understood. Around the time of gastrulation, most of the pair-rule genes undergo a transition from double-segmental stripes to single-segmental stripes. For *prd*, this happens by splitting of its early, broad pair-rule stripes. In contrast, *odd, runt* and *slp* show intercalation of 'secondary' stripes between their 'primary' 7-stripe patterns. Secondary stripes of *eve* also appear at gastrulation, but these 'minor' stripes (*Macdonald et al., 1986*) are extremely weak (usually undetectable in our fluorescent in situs), and not comparable to the rapidly developing segmental expression of *prd, odd, runt* and *slp*. Expression of *hairy* and *ftz* remains double segmental.

In some cases, discrete enhancer elements have been found that mediate just the secondary stripes (*Klingler et al., 1996*), while in other cases, all 14 segmental stripes are likely to be regulated coordinately (*Fujioka et al., 1995*). In certain cases, non-additive interactions between enhancers play a role in generating the segmental pattern (*Prazak et al., 2010*; *Gutjahr et al., 1994*). The functional significance of the late patterns is not always clear, since they are usually not reflected in pair-rule gene mutant cuticle phenotypes (*Kilchherr et al., 1986*; *Coulter et al., 1990*).

In the remainder of this paper, we investigate the nature and causes of the pattern transitions that occur between the end of phase 2 and the beginning of phase 3. A detailed analysis of the timing and dynamics of pair-rule gene expression during phase 2 will be covered elsewhere.

## Frequency-doubling of different pair-rule gene expression patterns is almost simultaneous, and coincides with segment-polarity gene activation

As noted above, four of the seven pair-rule genes undergo a transition from double-segment periodicity to regular single-segment periodicity at the end of cellularisation (*Figure 3*). These striking pattern changes could be caused simply by feedback interactions within the pair-rule and segment-polarity gene networks. Alternatively, they could be precipitated by some extrinsic temporal signal (or signals).

Comparing between genes, we find that the pattern changes develop almost simultaneously (*Figure 4*; *Figure 4—figure supplement 1*), although there are slight differences in the times at which the first signs of frequency-doubling become detectable. (The *prd* trunk stripes split just before the *odd* secondary stripes start to appear, while the secondary stripes of *slp* and *runt* appear just after). These events appear to be spatiotemporally modulated: they show a short but noticeable AP time lag, and also a DV pattern – frequency-doubling occurs first mid-laterally, and generally does not extend across the dorsal midline. In addition, the secondary stripes of *slp* are not expressed in the mesoderm, while the ventral expression of *odd* secondary stripes is only weak.

We also investigated the timing of the frequency-doubling events relative to the appearance of expression of the segment-polarity genes *en, gooseberry (gsb)* and *wingless (wg)* (*Figure 4*; *Figure 4—figure supplement 2*). We find that the spatiotemporal pattern of segment-polarity gene activation coincides closely with that of pair-rule frequency-doubling – starting at the beginning of phase 3, and rapidly progressing over the course of gastrulation. Only around 20 min separate a late stage 5 embryo (with double-segment periodicity of pair-rule gene expression and no segment-polarity gene expression) from a late stage 7 embryo (with regular segmental expression of both pair-rule genes and segment-polarity genes) (*Campos-Ortega and Hartenstein, 1985*).

We can make three conclusions from the timing of these events. First, segment-polarity gene expression cannot be precipitating the frequency-doubling of pair-rule gene expression, because frequency-doubling occurs before segment-polarity proteins would have had time to be synthesised. Second, the late, segmental patterns of pair-rule gene expression do not play a role in regulating the initial expression of segment-polarity genes, because they are not reflected at the protein level until after segmental patterns of segment-polarity gene transcripts are observed. Third, the synchrony of pair-rule gene frequency-doubling and segment-polarity gene activation is consistent with co-regulation of these events by a single temporal signal.

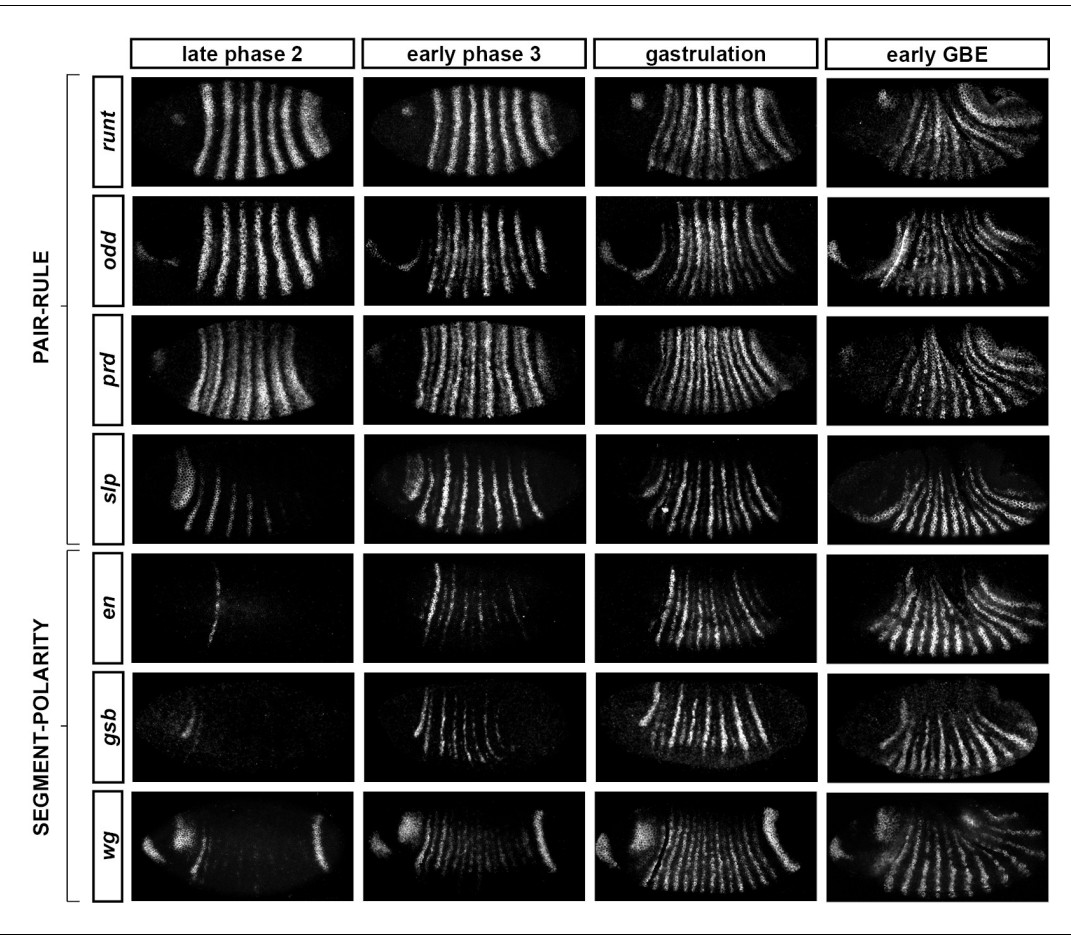

**Figure 4.** Frequency-doubling of pair-rule gene expression patterns is almost simultaneous and coincides with the first expression of the segment-polarity genes. Each row shows the expression of a particular pair-rule gene or segment-polarity gene, while each column represents a particular developmental timepoint. Late phase 2 and early phase 3 both correspond to late Bownes stage 5; gastrulation is Bownes stage 6, and early germband extension is Bownes stage 7 (*Bownes, 1975*; *Campos-Ortega and Hartenstein, 1985*). All panels show a lateral view, anterior left, dorsal top. GBE = germband extension. The figure represents about 20 min of development at 25°C.

The following figure supplements are available for figure 4:

**Figure supplement 1.** Relative expression of pair-rule genes during frequency-doubling.

**Figure supplement 2.** Relative expression of segment-polarity genes and pair-rule genes during frequency-doubling.

## The transition to single-segment periodicity is mediated by altered regulatory interactions

It is clear that a dramatic change overtakes pair-rule gene expression at gastrulation. For a given gene, an altered pattern of transcriptional output could result from an altered spatial pattern of regulatory inputs, or, alternatively, altered regulatory logic. Pair-rule proteins provide most of the spatial regulatory input for pair-rule gene expression at both phase 2 and phase 3. Therefore, the fact that the distributions of pair-rule proteins are very similar at the end of phase 2 and the beginning of phase 3 (*Pisarev et al., 2009*) suggests that it must be the 'input-output functions' of pair-rule gene transcription that change to bring about the new expression patterns.

For example, consider the relative expression patterns of *prd* and *odd* (*Figure 5*). There is abundant experimental evidence that the splitting of the *prd* stripes is caused by direct repression by Odd protein. The primary stripes of *odd* lie within the broad *prd* stripes, and the secondary interstripes that form within the *prd* stripes at gastrulation correspond precisely to those cells that express *odd* (*Figure 5D*). Furthermore, the *prd* stripes do not split in *odd* mutant embryos (*Baumgartner and Noll, 1990*; *Saulier-Le Dréan et al., 1998*), and *prd* expression is largely repressed by ectopically expressed Odd protein (*Saulier-Le Dréan et al., 1998*; *Goldstein et al., 2005*).

However, prior to *prd* stripe splitting, *prd* and *odd* are co-expressed in the same cells, with no sign that *prd* is sensitive to repression by Odd (*Figure 5C*). Because *prd* expression begins at a time when Odd protein is already present (*Pisarev et al., 2009*), this co-expression cannot be explained by protein synthesis delays. We therefore infer that Odd only becomes a repressor of *prd* at gastrulation, consistent with previous observations that aspects of Odd regulatory activity are temporally restricted (*Saulier-Le Dréan et al., 1998*).

This apparent temporal switch in the regulatory function of Odd is not unique. We have carefully examined pair-rule gene stripe phasings just before and just after the double-segment to single-segment transition, and find that these patterns do indeed indicate significant changes to the control logic of multiple pair-rule genes. The results of this analysis are presented in Appendix 1. In summary, a number of regulatory interactions seem to disappear at the beginning of phase 3: repression of *odd* by Hairy, repression of *odd* by Eve, and repression of *slp* by Runt. These regulatory interactions are replaced by a number of new interactions: repression of *prd* by Odd, repression of *odd* by Runt, repression of *runt* by Eve and repression of *slp* by Ftz. At the same time that these regulatory changes are observed, new elements for *eve* and *runt* turn on and various segment-polarity genes start to be expressed.

The outcome of all these regulatory changes is a coordinated transition to single-segment periodicity. We have schematised this transition in *Figure 6*. Our diagrams are in broad agreement with the interpretation of Jaynes and Fujioka (*Jaynes and Fujioka, 2004*), although we characterise the

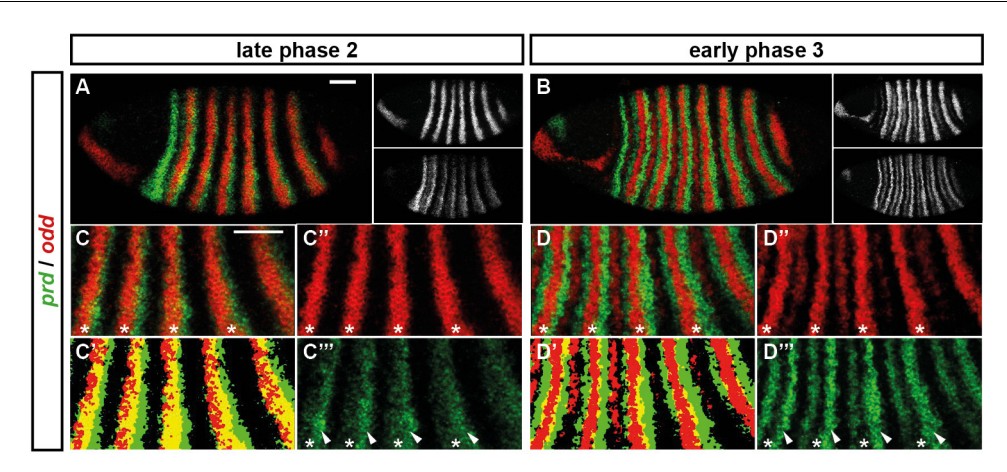

**Figure 5.** Odd does not repress *prd* transcription until phase 3. Relative expression of *prd* and *odd* is shown in a late phase 2 embryo (just prior to frequency doubling) and an early phase 3 embryo (showing the first signs of frequency doubling). (A, B) Whole embryos, lateral view, anterior left, dorsal top. Individual channels are shown to the right of each double-channel image, in the same vertical order as the panel label. (C, D) Blow-ups of expression in stripes 2–6; asterisks mark the location of *odd* primary stripes. Thresholded images (C', D') highlight regions of overlapping expression (yellow pixels). Considerable overlap between *prd* and *odd* expression is observed at phase 2 but not at phase 3. Note that the *prd* expression pattern is the combined result of initially broad stripes of medium intensity, and intense two-cell wide 'P' stripes overlapping the posterior of each of the broad stripes (arrowheads in C''', D'''). The two sets of stripes are mediated by separate stretches of DNA (*Gutjahr et al., 1994*), and must be regulated differently, since the 'P' stripes remain insensitive to ectopic Odd even during phase 3 (*Saulier-Le Dréan et al., 1998*; *Goldstein et al., 2005*). Scale bars = 50 µm.

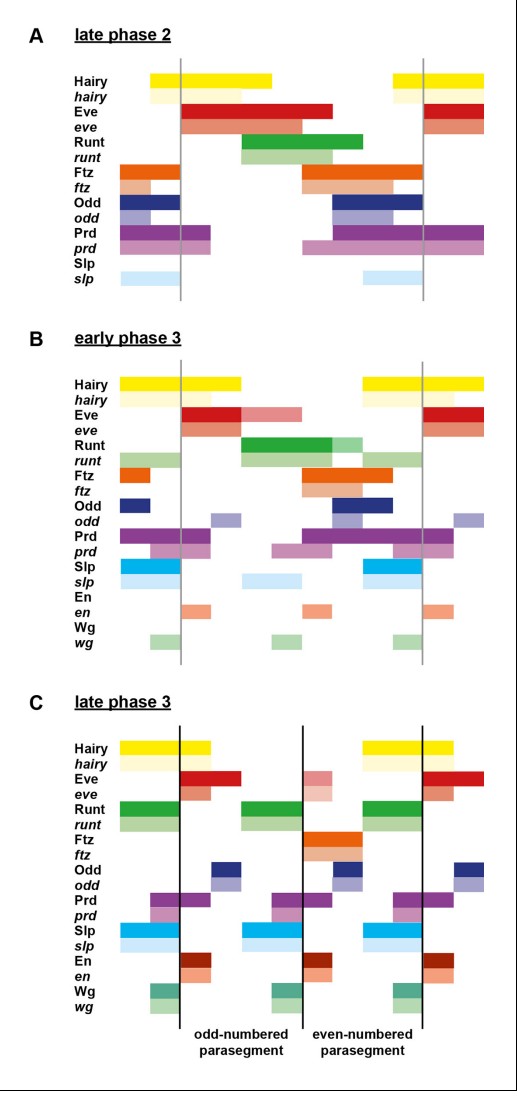

**Figure 6.** Schematic diagram of the transition to single-segment periodicity. Schematic diagram showing segmentation gene expression at late phase 2 (A), early phase 3 (B) and late phase 3 (C). The horizontal axis represents an idealised portion of the AP axis (~12 nuclei across). The grey vertical lines in (A, B) demarcate a double parasegment repeat (~8 nuclei across), while black lines in (C) indicate future parasegment boundaries. The patterns of protein expression (intense colours) and transcript expression (paler colours) of the pair-rule genes are shown at each timepoint. Those of the segment-polarity genes *en* and *wg* are additionally shown at the later timepoints. Transcript distributions were inferred from our double in situ data, while pair-rule protein distributions were inferred mainly from triple antibody stains in the FlyEx database (*Pisarev et al., 2009*). Additional protein expression information for late phase 3 (equivalent to the onset of germband extension) was gathered from published descriptions (*Frasch et al., 1987*; *DiNardo et al., 1985*; *van den Heuvel et al., 1989*;

*Figure 6 continued on next page*

process in greater temporal detail and distinguish between transcript and protein distributions at each timepoint.

## A candidate temporal signal: Odd-paired

Having identified the regulatory changes detailed above, we wanted to know how they are made to happen in the embryo. Because they all occur within a very short time window (*Figure 4*), they could potentially all be co-regulated by a single temporal signal that would instruct a regulatory switch. We reasoned that if this hypothetical signal were absent, the regulatory changes would not happen. This would result in a mutant phenotype in which frequency-doubling events do not occur, and segment-polarity expression is delayed.

We then realised that this hypothetical phenotype was consistent with descriptions of segmentation gene expression in mutants of the non-canonical pair-rule gene, *odd-paired (opa)* (*Benedyk et al., 1994*). This gene is required for the splitting of the *prd* stripes and the appearance of the secondary stripes of *odd* and *slp* (*Baumgartner and Noll, 1990*; *Benedyk et al., 1994*; *Swantek and Gergen, 2004*). It is also required for the late expression of *runt* (*Klingler and Gergen, 1993*), and for the timely expression of *en* and *wg* (*Benedyk et al., 1994*).

The *opa* locus was originally isolated on account of its cuticle phenotype, in which odd-numbered segments (corresponding to even-numbered parasegments) are lost (*Jürgens et al., 1984*). For many years afterwards, *opa* was assumed to be expressed in a periodic pattern of double-segment periodicity similar to the other seven pair-rule genes (for example, see *Coulter and Wieschaus, 1988*; *Ingham and Baker, 1988*; *Weir et al., 1988*; *Baumgartner and Noll, 1990*; *Lacalli, 1990*). When *opa*, which codes for a zinc finger transcription factor, was finally cloned, it was found – surprisingly – to be expressed uniformly throughout the trunk (*Benedyk et al., 1994*). Presumed to be therefore uninstructive for spatial patterning, it has received little attention in the context of segmentation since. However, we realised that Opa could still be playing an important role in spatial patterning. By providing temporal information that would act combinatorially with the spatial information carried by the canonical pair-rule genes, Opa might permit individual pair-rule genes to carry out different patterning roles at different points in time.

*Figure 6 continued*

*Gutjahr et al., 1993*; *Lawrence and Johnston, 1989*; *Carroll et al., 1988*). Fading expression of Eve and Runt is represented by lighter red and green sections in (**B**). The transient 'minor' stripes of Eve are represented by faint red in (**C**). Note that this diagram does not capture the graded nature of pair-rule protein distributions during cellularisation.

## Expression of *opa* spatiotemporally correlates with patterning events

We examined *opa* expression relative to other segmentation genes, and found an interesting correlation with the spatiotemporal pattern of segmentation (*Figure 7*). As previously reported (*Benedyk et al., 1994*), the earliest expression of *opa* is in a band at the anterior of the trunk, which we find corresponds quite closely with the head stripe of *prd* (data not shown). Expression in the rest of the trunk quickly follows, and persists until germband extension, at which point expression becomes segmentally modulated (*Figure 7I*).

*opa* begins to be transcribed throughout the trunk during phase 1, before regular patterns of pair-rule gene expression emerge (*Figure 7A*). The sharp posterior border of the *opa* domain at first lies just anterior to *odd* stripe 7 (*Figure 7B–E*), but gradually shifts posteriorly over the course of gastrulation to encompass it (*Figure 7F–H*). Notably, *odd* stripe 7 is the last of the primary pair-rule gene stripes to appear, and segmentation of this posterior region of the embryo appears to be significantly delayed relative to the rest of the trunk (*Kuhn et al., 2000*).

The timing of *opa* transcription has been shown to rely on nuclear / cytoplasmic ratio (*Lu et al., 2009*), and begins relatively early during cellularisation. However, it takes a while for the *opa* expression domain to reach full intensity. Unlike the periodically expressed pair-rule genes, which have

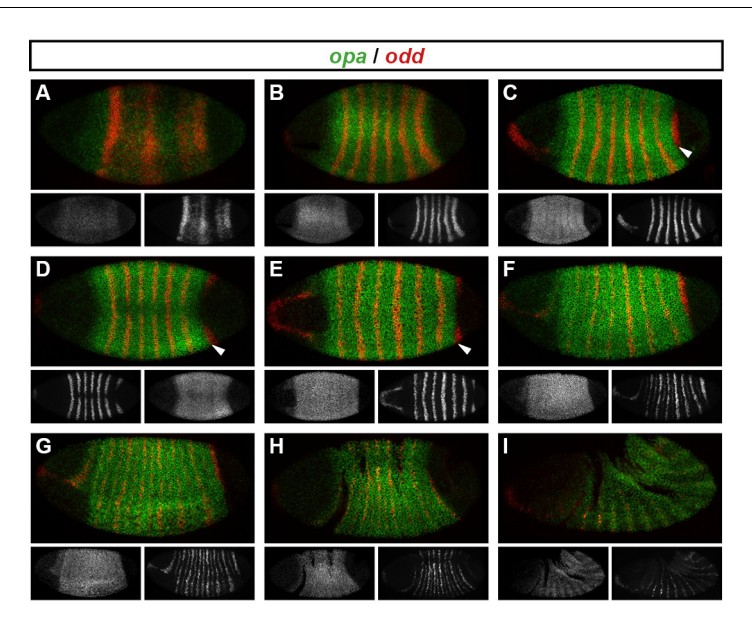

**Figure 7.** Spatiotemporal expression of *opa* relative to *odd*. Expression of *opa* relative to *odd* from early cellularisation until mid germband extension. (**A**) phase 1, lateral view; (**B**) early phase 2; (**C–E**) late phase 2; (**F**) early phase 3; (**G**, **H**) gastrulation; (**I**) early germband extension. Anterior left; (**A**, **B**, **C**, **F**, **I**) lateral views; (**D**) dorsal view; (**E**) ventral view; (**G**) ventrolateral view; (**H**) dorsolateral view. Single-channel images are shown in greyscale below each double-channel image (*opa* on the left, *odd* on the right). Arrowheads in (**C–E**) point to the new appearance of *odd* stripe 7, which abuts the posterior boundary of the *opa* domain. Note that *odd* stripe 7 is incomplete both dorsally (**D**) and ventrally (**E**). By gastrulation, *opa* expression has posteriorly expanded to cover *odd* stripe 7 (**G**, **H**). *opa* expression becomes segmentally modulated during germband extension (**I**).

The following figure supplement is available for figure 7:

**Figure supplement 1.** The cellular localisation of *opa* transcripts changes over the course of segmentation.

compact transcription units (all <3.5 kb, FlyBase) consistent with rapid protein synthesis, the *opa* transcription unit is large (~17 kb, FlyBase), owing mainly to a large intron. Accordingly, during most of cellularisation, we observe a punctate distribution of *opa*, suggestive of nascent transcripts located within nuclei (*Figure 7—figure supplement 1*). Unfortunately, the available polyclonal anti-body against Opa (*Benedyk et al., 1994*) did not work well in our hands, so we have not been able to determine precisely what time Opa protein first appears in blastoderm nuclei. However, Opa protein levels have been reported to peak at late cellularisation and into gastrulation (*Benedyk et al., 1994*), corresponding to the time at which we observe regulatory changes in the embryo, and consistent with our hypothesised role of Opa as a temporal signal.

## *opa* mutant embryos do not transition to single-segment periodicity at gastrulation

If our hypothesised role for Opa is correct, patterning of the pair-rule genes should progress normally in *opa* mutant embryos up until the beginning of phase 3, but not undergo the dramatic pattern changes observed at this time in wild-type. Instead, we would expect that the double-segmental stripes would persist unaltered, at least while the activators of phase 2 expression remain present. The pair-rule gene expression patterns that have been described previously in *opa* mutant embryos (see above) seem consistent with this prediction; however, we wanted to characterise the *opa* mutant phenotype in more detail to be sure.

Throughout cellularisation, we find that pair-rule gene expression is relatively normal in *opa* mutant embryos (*Figure 8*; *Figure 8—figure supplement 1*), consistent with our hypothesis that Opa function is largely absent from wild-type embryos during these stages. During late phase 2, we observe only minor quantitative changes to the pair-rule stripes: the *odd* primary stripes seem wider than normal, the *prd* primary stripes seem more intense than normal, and the *slp* primary stripes – which normally appear at the very end of phase 2 – are weakened and delayed.

In contrast, pair-rule gene expression becomes dramatically different from wild-type at gastrulation (*Figure 8*; *Figure 8—figure supplement 2*). Most notably, the transition from double-segment to single-segment periodicity is not observed for any pair-rule gene – for example, the secondary stripes of *odd* and *slp* do not appear, and the *prd* stripes do not split. In addition, the primary stripes of *ftz* and *odd* remain broad, similar to their expression during phase 2, rather than narrowing from the posterior as in wild-type.

Not all the pair-rule genes remain expressed in pair-rule stripes. Except for stripes 6 and 7, the *runt* primary stripes are lost, replaced by fairly ubiquitous weak expression which nevertheless retains a double-segmental modulation. *eve* expression – which has not to our knowledge been previously characterised in *opa* mutant embryos – fades from stripes 3–7, with no sign of the sharpened 'late' expression normally activated in the anteriors of the early stripes (*Figure 8—figure supplement 5*). *hairy* expression fades much as it does in wild-type, except that there is reduced separation between certain pairs of stripes.

The expression patterns seen at gastrulation persist largely unaltered into germband extension (*Figure 8*; *Figure 8—figure supplement 3*), with the exception that the *slp* stripes expand anteriorly, overlapping the domains of *odd* expression. The persistence of the intense *prd* stripes (which overlap those of *ftz*, *odd* and *slp*, and remain strongly expressed throughout germband extension) is especially notable given that *prd* expression fades from wild-type embryos soon after gastrulation.

## Opa accounts for the regulatory changes observed at gastrulation

In summary, in *opa* mutant embryos *odd, prd* and *slp* remain expressed in pair-rule patterns after gastrulation, while expression of *eve* and *runt* is largely lost (schematised in *Figure 8—figure supplement 4*). The aberrant expression patterns of *odd, prd* and *slp* appear to directly reflect an absence of the regulatory changes normally observed in wild-type at phase 3. For example, the altered *prd* pattern is consistent with Odd failing to repress *prd*, indicating that Odd only acts as a repressor of *prd* in combination with Opa. Similarly, the expression pattern of *slp* is consistent with continued repression from Runt (a phase 2 interaction) and an absence of repression from Ftz (a phase 3 interaction), indicating that Runt only represses *slp* in the absence of Opa, while the opposite is true for Ftz. In Appendix 2, we demonstrate how an Opa-dependent switch from repression of *odd* by Eve (phase 2) to repression of *odd* by Runt (phase 3) is important for the precise

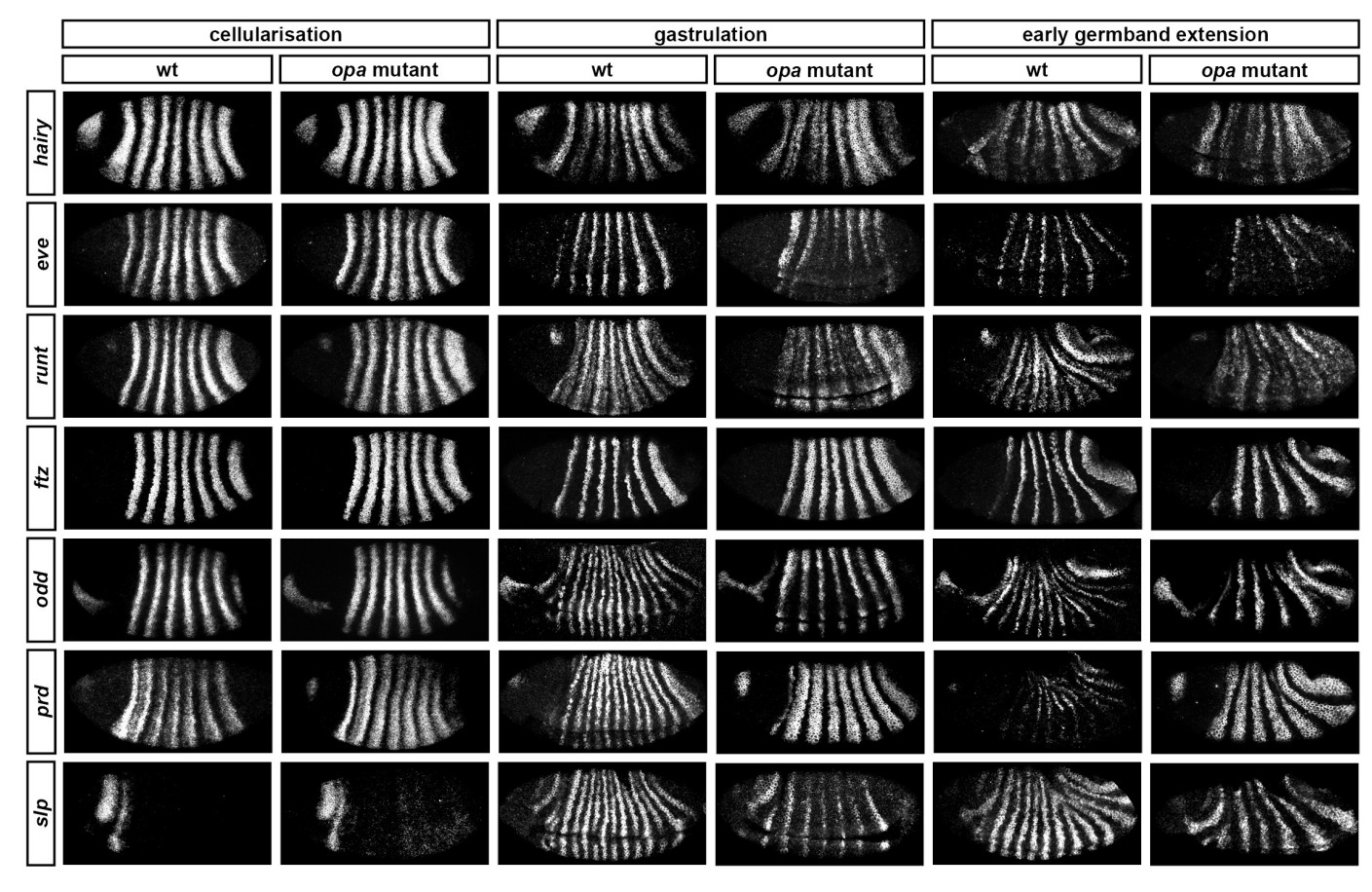

**Figure 8.** Pair-rule gene expression is perturbed from gastrulation onwards in *opa* mutant embryos. Pair-rule gene expression in wild-type and *opa* mutant embryos at late cellularisation, late gastrulation, and early germband extension. During cellularisation, pair-rule gene expression in *opa* mutant embryos is very similar to wild-type. Expression from gastrulation onwards is severely abnormal; in particular, note that single-segment patterns do not emerge. All panels show a lateral view, anterior left, dorsal top.

The following figure supplements are available for figure 8:

**Figure supplement 1.** Pair-rule gene expression in *opa* mutant embryos at cellularisation.

**Figure supplement 2.** Pair-rule gene expression in *opa* mutant embryos at gastrulation.

**Figure supplement 3.** Pair-rule gene expression in *opa* mutant embryos at early germband extension.

**Figure supplement 4.** The transition to single-segment periodicity does not occur in *opa* mutant embryos.

**Figure supplement 5.** Opa activates the *eve* "'late' element.

**Figure supplement 6.** 'Late' *eve* expression is observed in cells that do not express *prd*.

positioning of the anterior borders of the *odd* primary stripes, in addition to being necessary for the emergence of the *odd* secondary stripes.

The loss of *eve* and *runt* expression in *opa* mutant embryos indicates first that the activators that drive expression of *eve* and *runt* during phase 2 do not persist in the embryo after the end of cellularisation, and second that the expression of these genes during phase 3 is activated by the new appearance of Opa. The inference of different activators at phase 2 and phase 3 is not too surprising

for *eve*, which has phase 2 expression driven by stripe-specific elements and phase 3 expression driven by a separate 'late' element (see below). Indeed, expression of stripe-specific elements is known to fade away at gastrulation, as seen for endogenous expression of *hairy* (*Ingham et al., 1985*; *Figure 8*), for stripe-specific reporter elements of *eve* (*Bothma et al., 2014*), or for transgenic embryos lacking *eve* late element expression (*Fujioka et al., 1995*). However, a single stretch of DNA drives *runt* primary stripe expression at both phase 2 and phase 3 (*Klingler et al., 1996*). This suggests that the organisation and regulatory logic of this element may be complex, as it is evidently activated by different factors at different times.

Opa is also likely to contribute to the activation of the *slp* primary stripes, explaining why they are initially weaker than normal in *opa* mutant embryos. (However, in this case Opa must act semi-redundantly with other activators, in contrast to its effects on *eve* and *runt*.) A resulting delay in the appearance of Slp protein in *opa* mutant embryos could account for the broadened stripes of *ftz* and *odd*, which normally narrow during phase 3 in response to repression from Slp at the posterior. Alternatively, these regulatory functions of Slp could themselves be directly Opa-dependent.

## Opa activates the *eve* 'late' element

Our discovery that Opa was required for late *eve* expression (*Figure 8—figure supplement 5*) was surprising, because the enhancer element responsible for this expression has been studied in detail (*Goto et al., 1989*; *Harding et al., 1989*; *Jiang et al., 1991*; *Fujioka et al., 1996*; *Sackerson et al., 1999*), and Opa has not previously been implicated in its regulation. The *eve* 'late' element is sometimes referred to as the *eve* 'autoregulatory' element, because expression from it is lost in *eve* mutant embryos (*Harding et al., 1989*; *Jiang et al., 1991*). However, the observed 'autoregulation' appears to be indirect (*Goto et al., 1989*; *Manoukian and Krause, 1992*; *Fujioka et al., 1995*; *Sackerson et al., 1999*). Instead of being directly activated by Eve, the element mediates regulatory inputs from repressors such as Runt and Slp, which are ectopically expressed in *eve* mutant embryos (*Vavra and Carroll, 1989*; *Klingler and Gergen, 1993*; *Riechmann et al., 1997*; *Jaynes and Fujioka, 2004*). The element is thought to be directly activated by Prd, and functional Prd-binding sites have been demonstrated within it (*Fujioka et al., 1996*). However, while Prd protein appears at roughly the right time to activate the *eve* late element (*Pisarev et al., 2009*), activation by Prd cannot explain all the expression generated from this element, because during early phase 3 it drives expression in many cells that do not express *prd* (*Figure 8—figure supplement 6*).

Instead, it seems that the *eve* late element is directly activated by Opa. The lack of late *eve* expression in *opa* mutant embryos cannot be explained by the ectopic expression of repressive inputs, since none of *runt, odd* or *slp* are ectopically expressed in the domains where *eve* late element expression would normally be seen (*Figure 8*; *Figure 8—figure supplement 4*). Furthermore, the total loss of *eve* expression in certain stripes despite the presence of appropriately positioned *prd* expression indicates that Prd alone is not sufficient to drive strong *eve* expression. Activation of *en* expression by Prd also requires the presence of Opa (*Benedyk et al., 1994*), suggesting that cooperative interactions between Prd and Opa might be common.

## Opa regulatory activity may be concentration-dependent

Not all the Opa-dependent expression pattern changes we identified through our analysis of *opa* mutant embryos happen at exactly the same time in wild-type embryos. Specifically, the splitting of the *prd* stripes and the appearance of the *slp* primary stripes occur a few minutes earlier than the other changes, such as the appearance of the secondary stripes of *odd* and *slp*, and the late expression of *eve*. If we assume that Opa concentration increases in the embryo over time as more protein is synthesised, these timing discrepancies could be explained by the former events being driven a lower level of Opa activity than required for the latter events.

In order to investigate this hypothesis, we examined pair-rule gene expression in mutants for a 'weak' allele of *opa* (*opa⁵*, also known as *opa¹³ᴰ⁹²*) which we presume to represent an *opa* hypomorph. Whereas mutants for the null allele we investigated (*opa⁸*, also known as *opa¹¹ᴾ³²*) develop cuticles with complete pairwise fusion of adjacent denticle belts, mutants for *opa⁵* develop less severe patterning defects where denticle belts remain separate or only partially fuse (*Baumgartner et al., 1994*).

*Figure 9* compares expression patterns in *opa* hypomorphic embryos to both the wild-type and null situations. At cellularisation, expression patterns are similar for all three genotypes (data not shown). At gastrulation, expression patterns in the hypomorphic embryos tend to resemble those in the null embryos. However, there are two significant differences, corresponding to the two Opa-dependent patterning events that occur first in wild-type embryos. First, the *slp* primary stripes are expressed more strongly in the hypomorphic embryos than in the null embryos (although their appearance is still slightly delayed), and second, the *prd* stripes in the hypomorphic embryos show weak expression in the centre of the stripes (arrowheads in *Figure 9*), a situation intermediate between the wild-type situation of full splitting, and the null situation of completely uniform stripes. Later, during germband extension, expression patterns in the hypomorphic embryos diverge further from the null situation, with multiple genes exhibiting evidence of Opa-dependent regulation (arrowheads in *Figure 9*). For example, the *prd* stripes fully split, some evidence of *odd* and *slp* secondary stripes can be seen, and strong *runt* expression is reinitiated.

Together, this evidence suggests that different regulatory targets of Opa respond with differential sensitivity. As the level of Opa increases over time, 'sensitive' targets would show expression changes as soon as a low threshold of Opa activity was reached, whereas other targets would respond later, when a higher threshold was reached. In wild-type embryos, the low threshold events occur slightly earlier than the high threshold events, at late phase 2 rather than early phase 3. In *opa* hypomorphic embryos, in which the rate of increase in Opa activity would be slower, these events happen later but still in the same temporal sequence, with the low threshold events occurring at

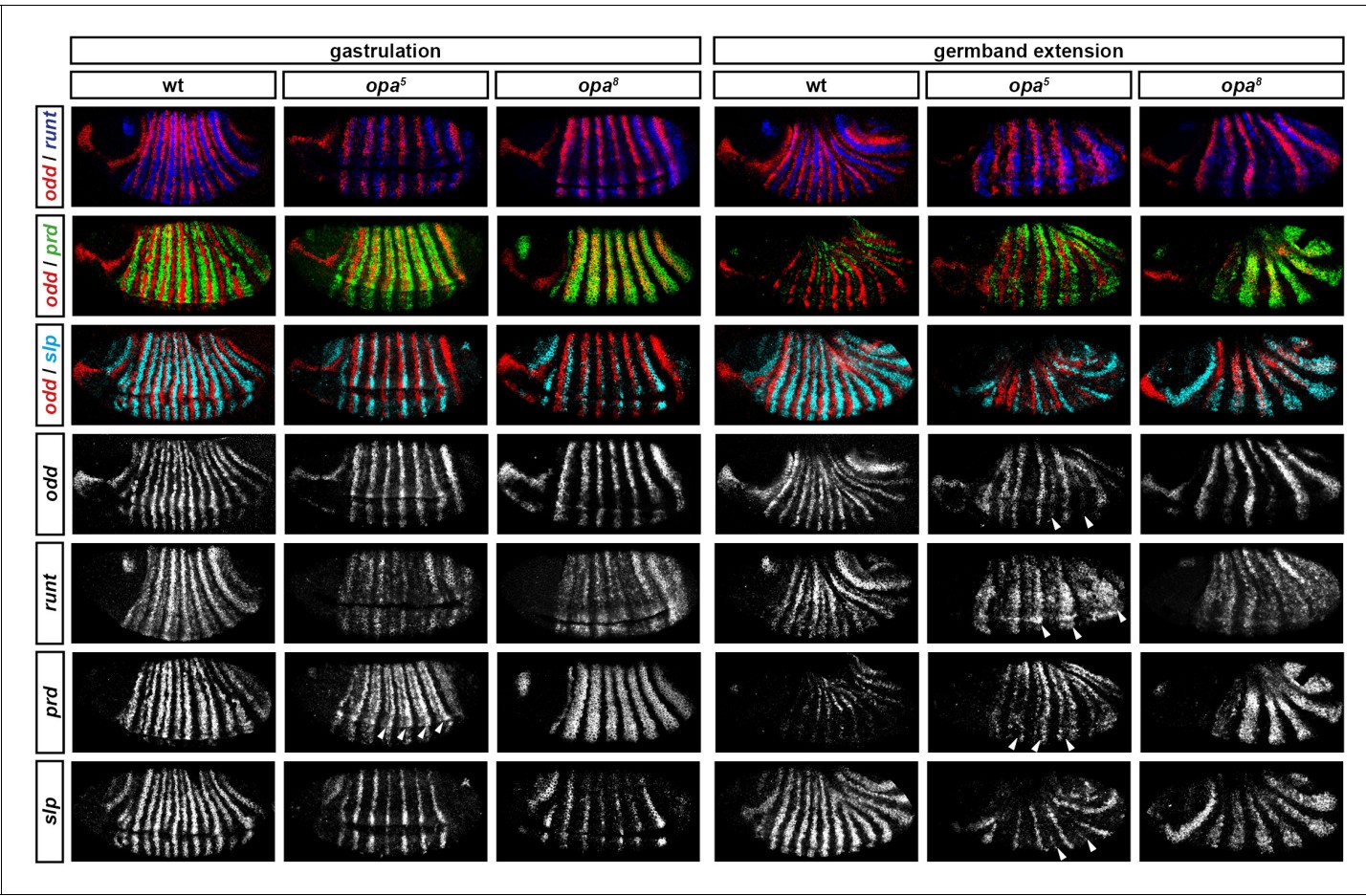

**Figure 9.** Opa-dependent expression pattern changes are delayed in *opa* hypomorphic embryos. Expression of selected pair-rule genes compared between embryos wild-type, hypomorphic (*opa⁵*), or null mutant (*opa⁸*) for *opa*. Arrowheads mark evidence of Opa-dependent regulatory interactions in *opa⁵* embryos (see text for details). All panels show a lateral view, anterior left, dorsal top.

gastrulation, and the high threshold events not detected until germband extension. In *opa* null embryos, both classes of events of course do not happen at all.

## *opa* mutant embryos fail to pattern the even-numbered parasegment boundaries

Explaining the aetiology of the *opa* pair-rule phenotype requires understanding why the loss of Opa activity results in the mispatterning of parasegment boundaries by segment-polarity genes. In wild-type embryos, *en* and *wg* are initially regulated cell-autonomously by pair-rule proteins (for example, see *Ingham and Baker, 1988*; *Weir et al., 1988*; *Manoukian and Krause, 1993*; *Mullen and DiNardo, 1995*). During germband extension, they become dependent on intercellular signalling for their continued expression, with the Wingless and Hedgehog signalling pathways forming a positive feedback loop that maintains each parasegment boundary (*DiNardo et al., 1988*, *1994*; *Perrimon, 1994*; *von Dassow et al., 2000*).

Expression of *en* and *wg* has previously been characterised in *opa* mutant embryos, demonstrating that the even-numbered parasegment boundaries fail to establish properly (*Benedyk et al., 1994*; *Ingham, 1986*; *DiNardo and O'Farrell, 1987*; see also *Figure 10—figure supplement 1*). To summarise, although their initial appearance is somewhat delayed, the even-numbered *wg* stripes (which normally contribute to the odd-numbered parasegment boundaries) and some of the even-numbered *en* stripes (which normally contribute to the even-numbered parasegment boundaries) become established in their normal locations by the beginning of the germband extension. Later on in germband extension, odd-numbered *en* stripes become established adjacent to the even-numbered *wg* stripes, leading to the formation of the odd-numbered parasegment boundaries. In contrast, the odd-numbered *wg* stripes never appear, the even-numbered *en* stripes eventually fade away, and the even-numbered parasegment boundaries are not established.

Our characterisation of pair-rule gene expression in *opa* mutant embryos enables us to make sense of these patterns. First, Opa appears to regulate *slp* and *wg* in a very similar way (*Figure 10—figure supplement 1*). The even-numbered *wg* stripes overlap with the primary stripes of *slp* and show the same expression delays in *opa* mutant embryos, while the odd-numbered *wg* stripes and the *slp* secondary stripes, which would normally be activated at the same time and in the same places, both fail to appear. Second, the activation of *en* by Prd seems to strictly require Opa activity, whereas the activation of *en* by Ftz does not. Therefore, while the odd-numbered (Prd-activated) *en* stripes are initially absent in *opa* mutant embryos, some of the even-numbered (Ftz-activated) stripes do appear, although this is compromised by ectopic expression of Odd (*Benedyk et al., 1994*, Appendix 2). Third, the capacity for a partially specified parasegment boundary to later recover depends on the presence of an appropriate segmental pattern of pair-rule gene expression, despite these patterns arising too late to regulate the initial expression of segment-polarity genes at gastrulation.

For example, the Slp stripes play an important segment-polarity role during germband extension, defining the posterior half of each parasegment. They repress *en* expression and are also necessary for the maintenance of *wg* expression (*Cadigan et al., 1994b*). In the case of the odd-numbered parasegment boundaries, *slp* and *wg* are properly patterned in *opa* mutant embryos, but the *en* stripes are absent (*Figure 10—figure supplement 1*). However, repressors of *en* such as Odd and Slp are not ectopically expressed in their place. Therefore, the odd-numbered *en* stripes are able to be later induced in their normal positions, presumably in response to Wg signalling coming from the Slp primary stripes, and therefore properly patterned boundaries eventually emerge. However, in the case of the even-numbered parasegment boundaries, while the *en* stripes are usually present in *opa* mutant embryos, both the *slp* stripes and the *wg* stripes are not (*Figure 10—figure supplement 1*). The absence of the Slp secondary stripes means that the cells anterior to the even-numbered En stripes are not competent to express *wg*. Hedgehog signalling from the even-numbered En stripes is therefore unable to induce the odd-numbered *wg* stripes, and consequently the boundaries do not recover.

Based on regulatory interactions analysed in Appendix 1 and Appendix 2, we present an updated model for how the even-numbered parasegment boundaries are specified in wild-type embryos (*Figure 10*). We propose that the spatial information directly responsible for patterning these boundaries derives from overlapping domains of Runt and Ftz activity (*Appendix 1—figure 3G,H*). Ftz and Runt combinatorially specify distinct expression domains of *slp*, *en* and *odd*, by way of late acting,

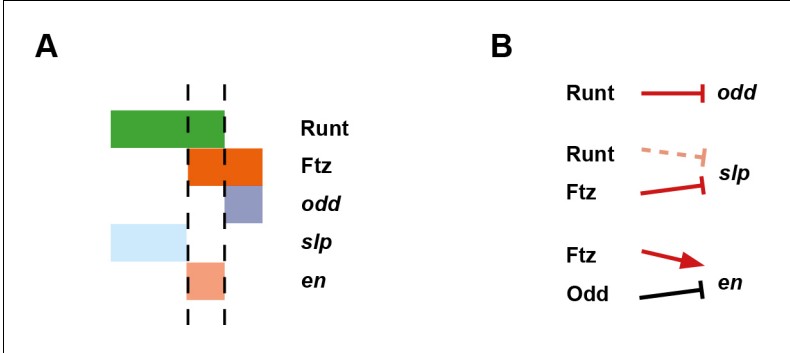

**Figure 10.** Model for the Opa-dependent patterning of the even-numbered parasegment boundaries. (**A**) Schematic showing the phasing of *odd, slp* and *en* relative to Runt and Ftz protein at phase 3. The horizontal axis represents part of a typical double-segment pattern repeat along the AP axis of the embryo (~4 nuclei across, centred on an even-numbered parasegment boundary). (**B**) Inferred regulatory interactions governing the expression of *odd, slp* and *en* at phase 3. Regular arrows represent activatory interactions; hammerhead arrows represent repressive interactions. Solid arrows represent interactions that are currently in operation; pale dashed arrows represent those that are not. Red arrows represent interactions that depend on the presence of Opa protein. Overlapping domains of Runt and Ftz expression (**A**) subdivide this region of the AP axis into three sections (black dashed lines). Opa-dependent repression restricts *odd* expression to the posterior section, resulting in offset anterior boundaries of Ftz and Odd activity (*Appendix 2—figure 1*; *Appendix 2—figure 1—figure supplement 2*). *slp* expression is restricted to the anterior section by the combination of Opa-dependent repression from Ftz and Opa-dependent de-repression from Runt (*Appendix 1—figure 2—figure supplement 1*). *en* is restricted to the central section by the combination of activation from Ftz (likely partially dependent on Opa), and repression by Odd. Later, mutual repression between *odd, slp* and *en* will maintain these distinct cell states. The even-numbered parasegment boundaries will form between the *en* and *slp* domains. Note that, in this model, Eve has no direct role in patterning these boundaries.

The following figure supplement is available for figure 10:

**Figure supplement 1.** Segment-polarity gene expression in *opa* mutant embryos.

Opa-dependent regulatory interactions. As described above, the loss of these interactions in *opa* mutant embryos results in mispatterning of *slp* and *odd* (*Figure 8—figure supplement 4*), which later has significant repercussions for segment-polarity gene expression.

## Discussion

### Opa alters the pair-rule network and temporally regulates segmentation

We have found that many regulatory interactions between the pair-rule genes are not constant over the course of *Drosophila* segmentation, but instead undergo coordinated changes at the end of cellularisation. We are not the first to notice that certain regulatory interactions do not apply to all stages of pair-rule gene expression (*Baumgartner and Noll, 1990*; *Manoukian and Krause, 1992*; *Manoukian and Krause, 1993*; *Fujioka et al., 1995*; *Saulier-Le Dréan et al., 1998*). However, cataloguing and analysing these changes for the whole pair-rule system led us to the realisation that they are almost simultaneous and mediate the transition from double-segment to single-segment periodicity. We propose that the product of the non-canonical pair-rule gene *opa* acts as a temporal signal that mediates these changes, and simultaneously activates the expression of segment-polarity genes. Analysis of pair-rule gene expression patterns in *opa* mutant embryos indicates that the phase-specific regulatory interactions we inferred from wild-type embryos appear to be modulated by Opa, and thus explained by the onset of Opa regulatory activity at gastrulation.

We argue that the pair-rule system should not be thought of as a static gene regulatory network, but rather two temporally and topologically distinct networks, each with its own dynamical behaviour

and consequent developmental patterning role. Pair-rule patterning can therefore be thought of as a two-stage process. In the absence of Opa, the early network patterns the template for the odd-numbered parasegment boundaries. Then, when Opa turns on, Opa-dependent regulatory interactions lead to the patterning of the even-numbered parasegment boundaries. Each stage of patterning uses the same source of positional information (the primary stripes of the pair-rule genes), but uses different sets of regulatory logic to exploit this information in different ways.

Opa thus plays a crucial timing role in segmentation, orchestrating the transition from pair-rule to segmental patterning. Notably, the role of Opa in activating the initial stages of segment-polarity gene expression demonstrates that segment-polarity gene expression is not simply induced by the emergence of an appropriate pattern of pair-rule proteins, as in textbook models of hierarchical gene regulation. The necessity for an additional signal had been surmised previously, based on the delayed appearance of odd-numbered *en* stripes in cells already expressing Eve and Prd (*Manoukian and Krause, 1993*).

Because correct segmentation depends upon the initial expression of segment-polarity genes being precisely positioned, it is imperative that a regular pair-rule pattern is present before the segment-polarity genes first turn on. Therefore, explicit temporal control of segment-polarity gene activation by Opa makes good sense from a patterning perspective. There are likely to be a number of analogous regulatory signals that provide extrinsic temporal information to the *Drosophila* segmentation cascade. For example, a ubiquitously expressed maternal protein, Tramtrack, represses pair-rule gene expression during early embryogenesis, effectively delaying pair-rule gene expression until appropriate patterns of gap gene expression have been established (*Harrison and Travers, 1990*; *Read et al., 1992*; *Brown and Wu, 1993*).

## What is the mechanism of Opa regulatory activity?

*opa* is the *Drosophila* ortholog of *zinc finger of the cerebellum (zic)* (*Aruga et al., 1994*). *zic* genes encode zinc finger transcription factors closely related to Gli proteins that have many important developmental roles.

In the *Drosophila* embryo, in addition to its role in segmentation, Opa is also involved in the formation of visceral mesoderm (*Cimbora and Sakonju, 1995*; *Schaub and Frasch, 2013*). Opa is later highly expressed in the larval and adult brain (FlyAtlas – *Chintapalli et al., 2007*) and is likely to be involved in neuronal differentiation (*Eroglu et al., 2014*). It is also involved in the regulation of adult head development (*Lee et al., 2007*).

The neuronal function is likely to reflect an ancestral role of Zic, as involvement of Zic genes in nervous system development and neuronal differentiation is pervasive throughout metazoans (*Layden et al., 2010*). Lineage-specific duplications have resulted in five *zic* genes in most vertebrate taxa, and seven in teleosts (*Aruga et al., 2006*; *Merzdorf, 2007*). While partial redundancy between these paralogs complicates the interpretation of mutant phenotypes, it is clear that in vertebrates Zic proteins play crucial roles in early embryonic patterning, neurogenesis, left-right asymmetry, neural crest formation, somite development, and cell proliferation (reviewed in *Merzdorf, 2007*; *Houtmeyers et al., 2013*).

Zic proteins have been shown to act both as classical DNA-binding transcription factors, and as cofactors that modulate the regulatory activity of other transcription factors via protein-protein interactions (reviewed in *Ali et al., 2012*; *Winata et al., 2015*). They show context-dependent activity and can both activate and repress transcription (*Yang et al., 2000*; *Salero et al., 2001*). In particular, they appear to be directly involved in the modulation and interpretation of Wnt and Hedgehog signalling (*Murgan et al., 2015*; *Pourebrahim et al., 2011*; *Fujimi et al., 2012*; *Koyabu et al., 2001*; *Chan et al., 2011*; *Quinn et al., 2012*). Finally, they may play a direct role in chromatin regulation (*Luo et al., 2015*).

The roles that Opa plays in the *Drosophila* segmentation network appear to be consistent with the mechanisms of Zic regulatory activity that have been characterised in vertebrates. Opa appears to transcriptionally activate a number of enhancers, including those driving late expression of *eve*, *runt* and *slp*. In the case of the *slp* enhancer, this has been verified experimentally (*Sen et al., 2010*). In other cases, the role of Opa is likely to be restricted to modulating the effect of other regulatory inputs, such as mediating the repressive effect of Odd on *prd* expression, or the activatory effect of Prd on *en* expression. It will be interesting to investigate the enhancers mediating late pair-rule

gene expression and early segment polarity gene expression, and to determine how Opa interacts with them to bring about these varied effects.

## Is Opa sufficient for the regulatory changes we observe at gastrulation?

Our data seem consistent with Opa being 'the' temporal signal that precipitates the 7 stripe to 14 stripe transition. However, it remains possible that Opa acts in conjunction with some other, as yet unidentified, temporally patterned factor, or has activity that is overridden during cellularisation by some maternal or zygotic factor that disappears at gastrulation. Indeed, combinatorial interactions with DV factors do seem likely to be playing a role in restricting the effects of Opa: despite the *opa* expression domain encircling the embryo, many Opa-dependent patterning events do not extend into the mesoderm or across the dorsal midline. Identification of these factors should yield insights into cross-talk between the AP and DV patterning systems of the *Drosophila* blastoderm.

The activity of Opa has previously been suggested to be concentration-dependent (*Swantek and Gergen, 2004*). By comparing pair-rule gene expression in embryos with varying levels of Opa activity, we found evidence that different enhancers show different sensitivity to the concentration of Opa in a nucleus, explaining why different Opa-dependent regulatory events happen at slightly different times in wild-type embryos.

One of the earliest responses to Opa regulatory activity is the appearance of the *slp* primary stripes. However, we note that while Opa may contribute to their timely activation, these stripes still emerge in *opa* null mutant embryos. This is not surprising, as the *slp* locus has been shown to possess multiple partially redundant regulatory elements driving spatially and temporally overlapping expression patterns (*Fujioka and Jaynes, 2012*). From our own observations, we have found multiple cases where mutation of a particular gene causes the *slp* primary stripes to be reduced in intensity, but not abolished (data not shown), suggesting that regulatory control of these expression domains is redundant at the *trans* level as well as at the *cis* level. Partially redundant enhancers that drive similar patterns, but are not necessarily subject to the same regulatory logic, appear to be very common for developmental transcription factors (*Cannavò et al., 2016*; *Hong et al., 2008*; *Perry et al., 2011*; *Staller et al., 2015*; *Wunderlich et al., 2015*).

## General regulatory principles of the pair-rule network

By carefully analysing pair-rule gene expression patterns in the light of the experimental literature (Appendix 1), we have clarified our understanding of the regulatory logic responsible for generating them. In particular, we propose significantly revised models for the patterning of *odd, slp* and *runt*. Because the structure of a regulatory network determines its dynamics, and its structure is determined by the control logic of its individual components, these subtleties are not merely developmental genetic stamp-collecting. Our reappraisal of the pair-rule gene network allows us to re-evaluate some long-held views about *Drosphila* blastoderm patterning.

First, pair-rule gene interactions are combinatorially regulated by an extrinsic source of temporal information, something not allowed for by textbook models of the *Drosophila* segmentation cascade. We have characterised the role of Opa during the 7 stripe to 14 stripe transition, but there may well be other such signals acting earlier or later. Indeed, context-dependent transcription factor activity appears to be very common (*Stampfel et al., 2015*).

Second, our updated model of the pair-rule network is in many ways simpler than previously thought. While we do introduce the complication of an Opa-dependent network topology, this effectively streamlines the sub-networks that operate early (phase 2) and late (phase 3). At any one time, each pair-rule gene is only regulated by two or three other pair-rule genes. We do not see strong evidence for combinatorial interactions between these inputs (*cf. DiNardo and O'Farrell, 1987*; *Baumgartner and Noll, 1990*; *Swantek and Gergen, 2004*). Instead, pair-rule gene regulatory logic seems invariably to consist of permissive activation by a broadly expressed factor (or factors) that is overridden by precisely positioned repressors (*Edgar et al., 1986*; *Weir et al., 1988*). This kind of regulation appears to typify other complex patterning systems, such as the vertebrate neural tube (*Briscoe and Small, 2015*).

Finally, pair-rule gene cross-regulation has traditionally been thought of as a mechanism to stabilise and refine stripe boundaries (e.g. *Edgar et al., 1989*; *Schroeder et al., 2011*). Consistent with

this function, as well as with the observed digitisation of gene expression observed at gastrulation (*Baumgartner and Noll, 1990*; *Pisarev et al., 2009*), we find that the late network contains a number of mutually repressive interactions (Eve/Runt, Eve/Slp, Ftz/Slp, Odd/Runt, Odd/Slp, and perhaps Odd/Prd). However, the early network does not appear to utilise these switch-like interactions, but is instead characterised by unidirectional repression (e.g. of *ftz* and *odd* by Eve and Hairy, or of *runt* by Odd). Interestingly, pair-rule gene expression during cellularisation has been observed to be unexpectedly dynamic (*Keränen et al., 2006*; *Surkova et al., 2008*), something that is notable given the oscillatory expression of pair-rule gene orthologs in short germ arthropods (*Sarrazin et al., 2012*; *El-Sherif et al., 2012*; *Brena and Akam, 2013*).

## Why do pair-rule genes show a late phase of expression?

We have shown that for the pair-rule genes, the transition to single-segment periodicity is mediated by substantial re-wiring of regulatory interactions. In addition, we have shown that this re-wiring is controlled by the same signal, Opa, that activates segment-polarity gene expression. We propose that Opa's effective role is to usher in a 'segment-polarity phase' of expression, in which both canonical segment-polarity factors, and erstwhile pair-rule factors, work together to define cell states. This hypothesis is consistent with the spatial patterns and regulatory logic of late pair-rule gene expression: most pair-rule genes become expressed in narrow segmental stripes, and partake in switch-like regulatory interactions consistent with segment-polarity roles. Furthermore, regulatory feedback from segment-polarity genes suggests the pair-rule genes become integrated into the segment-polarity network: for example, En protein is involved in patterning the late expression of *eve*, *odd*, *runt* and *slp* (*Harding et al., 1986*; *Mullen and DiNardo, 1995*; *Klingler and Gergen, 1993*; *Fujioka et al., 2012*).

However, the hypothesis that pair-rule factors perform segment-polarity roles is at odds with that fact that their mutants generally do not exhibit segment-polarity defects. We argue that this discrepancy can be resolved by accounting for partial redundancy with paralogous factors. For example, *slp* has a closely linked paralog, *slp2*, expressed almost identically, (*Grossniklaus et al., 1992*), and simultaneous disruption of both genes is required in order to reveal that the Slp stripes are a critical component of the segment-polarity network (*Cadigan et al., 1994a*; *1994b*). *prd* and *odd* also have paralogs, expressed in persistent segmental stripes coincident with their respective phase 3 expression patterns (*Baumgartner et al., 1987*; *Hart et al., 1996*). The *prd* paralog, *gsb*, gives a segment-polarity phenotype if mutated, but Prd and Gsb are able to substitute for each other if expressed under the control of the other gene's regulatory region (*Li and Noll, 1993*, *1994*; *Xue and Noll, 1996*), indicating that the same protein can fulfil both pair-rule and segment-polarity functions. Moreover, we have found that a deficiency removing both *odd* and its closely linked paralogs, *sob* and *drm*, gives a cuticle phenotype that shows segment-polarity defects corresponding to the locations of *odd* secondary stripes, in addition to the pair-rule defects characteristic of *odd* mutants (data not shown).

We envisage that ancestrally, the orthologs of *prd/gsb* and *odd/sob/drm* would have sequentially fulfilled both pair-rule and segment-polarity functions, employing different regulatory logic in each case. Later, these roles would have been divided between different paralogs, leaving the transient segmental patterns of *prd* and *odd* as evolutionary relics. Consistent with this hypothesis, the roles of *prd* and *gsb* seem to be fulfilled by a single co-ortholog, *pairberry1*, in grasshoppers, with a second gene, *pairberry2*, expressed redundantly (*Davis et al., 2001*).

Therefore, of the four pair-rule factors expressed in segmental patterns after gastrulation (Runt, Odd, Prd, Slp), at least three appear to have segment-polarity functions, although they may perform these roles only transiently before handing over the job to their paralogs. (No function has as yet been assigned to late Runt expression.) Because Hairy expression fades away after phase 2, that leaves only the functions of the late, double-segmental expression patterns of Eve and Ftz to be accounted for. Both these factors partake in the segment-polarity network by repressing *slp* and *wg* (*Fujioka et al., 2002*; *Swantek and Gergen, 2004*; *Copeland et al., 1996*). However, unlike canonical segment-polarity factors, their expression fades during germband extension. Functional equivalence with each other explains why, from a patterning perspective, they need not be expressed in every segment. Functional redundancy with En (*Fujioka et al., 2012*) explains why they need not be persistently expressed (indeed, En is the factor responsible for switching off late *eve* expression [*Harding et al., 1986*]). Given that *eve* shows a phase of single-segment periodicity in many pair-

rule insects (*Patel et al., 1994*; *Binner and Sander, 1997*; *Rosenberg et al., 2014*; *Mito et al., 2007*), (although not in *Bombyx mori* [*Nakao, 2010*]), it will be interesting to investigate whether a loss of regular segmental *eve* expression in the lineage leading to *Drosophila* is associated with changes to the roles of Ftz (and/or its cofactor, Ftz-F1) in segment patterning (*Heffer et al., 2013*, *2011*).

### Is the role of Opa conserved?

In light of our data, it will be interesting to characterise the role of Opa in other arthropod model organisms. The best studied short germ insect is the beetle *Tribolium castaneum*, which also exhibits pair-rule patterning. An RNAi screen of pair-rule gene orthologs reported no segmentation pheno-type for *opa* knock-down, and concluded that *opa* does not function as a pair-rule gene in *Tribolium* (*Choe et al., 2006*). However, the authors also state that *opa* knock-down caused high levels of lethality and most embryos did not complete development, indicating that this conclusion may be premature. In contrast to this study, iBeetle-Base (*Dönitz et al., 2015*) reports a segmentation phe-notype for *opa* knock-down (TC number: TC010234; iBeetle number: iB_04791). The affected cuticles show a reduced number of segments including the loss of the mesothorax (T2). This could indicate a pair-rule phenotype in which the even-numbered parasegment boundaries are lost, similar to the situation in *Drosophila.* If true, this suggests that at least some aspects of the role of Opa are conserved between long germ and short germ segmentation.

## Material and methods

### *Drosophila* mutants and husbandry

Wild-type embryos were Oregon-R. The pair-rule gene mutations used were *opa⁵* (Bloomington stock no. 5334), *opa⁸* (Bloomington stock no. 5335) and *ftz¹¹* (gift of Bénédicte Sanson). These muta-tions were balanced over *TM6C Sb Tb twi::lacZ* (Bloomington stock no. 7251) to allow homozygous mutant embryos to be easily distinguished. Two to four hours old embryos were collected on apple juice agar plates at 25°C, fixed in 4% paraformaldehyde (PFA) for 20 min according to standard pro-cedures, and stored at -20°C in methanol until required.

### RNA probes

Digoxigenin- (DIG) and fluorescein (FITC)-labelled riboprobes were generated using full-length pair-rule gene cDNAs from the *Drosophila* gene collection (*Stapleton et al., 2002*) and either DIG or fluorescein RNA labelling mix (Roche, Basel, Switzerland). The clones used were RE40955 (*hairy*); MIP30861 (*eve*); GH02614 (*runt*); IP01266 (*ftz*); GH22686 (*prd*); GH04704 (*slp*); LD30441 (*opa*); LD16125 (*en*); FI07617 (*gsb*); RE02607 (*wg*).

### Whole mount double fluorescent in situ hybridisation

Embryos were post-fixed in 4% PFA then washed in PBT (PBS with 0.1% Tween-20) prior to hybrid-isation. Hybridisation was performed at 56°C overnight in hybridisation buffer (50% formamide, 5x SSC, 5x Denhardt's solution, 100 µg/ml yeast tRNA, 2.5% w/v dextran sulfate, 0.1% Tween-20), with at least 1 hr of prehybridisation before introducing the probes. Embryos were simultaneously hybri-dised with one DIG probe and one FITC probe to different segmentation genes. Embryos from mutant crosses were additionally hybridised with a DIG probe to *lacZ*. Post-hybridisation washes were carried out as in *Lauter et al., 2011*. Embryos were then incubated in peroxidase-conjugated anti-FITC and alkaline phosphatase (AP)-conjugated anti-DIG antibodies (Roche, Basel, Switzerland) diluted 1:4000. Tyramide biotin amplification (TSA biotin kit, Perkin Elmer, Waltham, MA) followed by incubation in streptavidin Alexa Fluor 488 conjugate (ThermoFisher Scientific, Waltham, MA) was used to visualise the peroxidase signal. A Fast Red reaction (Fast Red tablets, Kem-En-Tec Diagnos-tics, Taastrup, Denmark) was subsequently used to visualise the AP signal. Embryos were mounted in ProLong Diamond Antifade Mountant (ThermoFisher Scientific) before imaging.

### Microscopy and image analysis

Embryos were imaged on a Leica SP5 Upright confocal microscope, using a 20x objective. For each pairwise combination of probes, a slide of ~100 embryos was visually examined, and around 20

images taken for further analysis. Occasional embryos with severe patterning abnormalities were discounted from analysis. Minor brightness and contrast adjustments were carried out using Fiji (*Schindelin et al., 2012*, *2012*). Thresholded images were produced using the 'Make Binary' option in Fiji. Our full wild-type dataset of over 600 double channel confocal images is available from the Dryad Digital Repository (*Clark and Akam, 2016*).

## Acknowledgements

The authors would like to thank all members of the Akam, Weil and Skaer groups, and the Department of Zoology imaging facility. Jack Green and Olivia Tidswell read and commented on the manuscript. Mutants were obtained from the Bloomington Stock Centre, and cDNA clones from the *Drosophila* Genomics Resource Centre. The *lacZ* cDNA was a gift from Nan Hu. This work was supported by a BBSRC PhD studentship to Erik Clark.

## Additional information

### Funding

| Funder | Grant reference number | Author |
| --- | --- | --- |
| Biotechnology and Biological Sciences Research Council | PhD Studentship | Erik Clark |

The funders had no role in study design, data collection and interpretation, or the decision to submit the work for publication.

### Author contributions

EC, Conception and design, Acquisition of data, Analysis and interpretation of data, Drafting or revising the article; MA, Conception and design, Drafting or revising the article

### Author ORCIDs

Erik Clark, http://orcid.org/0000-0002-5588-796X
Michael Akam, http://orcid.org/0000-0003-0063-2297

## Additional files

### Major datasets

The following dataset was generated:

| Author(s) | Year | Dataset title | Dataset URL | Database, license, and accessibility information |
| --- | --- | --- | --- | --- |
| Erik Clark, Michael Akam | 2016 | Data from: Odd-paired controls frequency doubling in Drosophila segmentation by altering the pair-rule gene regulatory network | http://dx.doi.org/10.5061/dryad.cg35k | Available at Dryad Digital Repository under a CC0 Public Domain Dedication |

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

## Appendix 1

### Regulatory changes between phase 2 and phase 3

In the main text, we presented phenomenological evidence for a change to the regulatory effect of Odd protein on *prd* transcription at the transition between phase 2 and phase 3. In this Appendix, we analyse the expression of the other six pair-rule genes before and after this transition, with particular focus on the inferred regulatory changes involved in mediating the altered expression patterns of *odd, slp, runt* and *eve*. Note that throughout what follows, italicised names (e.g. *eve*) are used to refer to genes and to the distributions of their transcript, whereas capitalised plain text (e.g. Eve) is used to refer to proteins and their distributions. Note also that primary pair-rule stripes shift anteriorly over the course of cellularisation (*Surkova et al., 2008*), and protein distributions lag slightly behind transcript distributions due to time delays inherent in protein synthesis and decay. This means that slight gaps tend to be present between the anterior border of a stripe and the transcripts of its anterior repressor (e.g. *Appendix 1—figure 1A*, *Appendix 1—figure 2C*), whereas slight overlaps may be seen between the posterior border of a stripe and the transcripts of its posterior repressor (e.g. *Appendix 1—figure 1C*, *Appendix 1—figure 3C*).

### *odd-skipped* (*Appendix 1—figure 1*; *Appendix 1—figure 1—figure supplement 1*)

During phase 2, the primary stripes of *odd* have anterior boundaries defined by repression by Eve, and posterior boundaries defined by repression by Hairy (*Manoukian and Krause, 1992*; *Jiménez et al., 1996*; *Appendix 1—figure 1A,C*). The primary stripes of *odd* narrow during phase 3, mainly from the posterior, and secondary stripes intercalate between them. It is not known whether all components of the single-segmental pattern observed at phase 3 are driven by a single enhancer, but we think it likely. The following analysis assumes that primary and secondary stripes of *odd* are governed by identical regulatory logic during phase 3.

The secondary stripes arise within cells expressing both Eve and Hairy (*Appendix 1—figure 1B,D*), indicating that repression of *odd* by these proteins is restricted to phase 2. A loss of repression by Hairy during phase 3 is also supported by increased overlaps between *hairy* and the *odd* primary stripes (*Appendix 1—figure 1D*). The posterior boundaries of the *odd* secondary stripes appear to be defined by repression by Runt. In wild-type embryos, these boundaries precisely abut the anterior boundaries of the *runt* primary stripes (*Appendix 1—figure 1F*), whereas in *runt* mutant embryos they expand posteriorly (*Jaynes and Fujioka, 2004*). However, *odd* is evidently not repressed by Runt during phase 2, because the *odd* primary stripes overlap with the posterior of the *runt* stripes (*Appendix 1—figure 1E*). The anterior boundaries of the *odd* secondary stripes appear to be defined by repression from Prd (*Figure 5D*), consistent with the observation that these stripes expand anteriorly in *prd* mutant embryos (*Mullen and DiNardo, 1995*). Since the *odd* primary stripes overlap with *prd* expression during phase 2 (*Figure 5C*), it is possible that repression of *odd* by Prd is restricted to phase 3. However, Prd protein appears relatively late during phase 2 (*Pisarev et al., 2009*), and Prd protein degradation is upregulated specifically in the region of the *odd* primary stripes (*Raj et al., 2000*), suggesting that Prd would have little effect on *odd* expression during phase 2 either way.

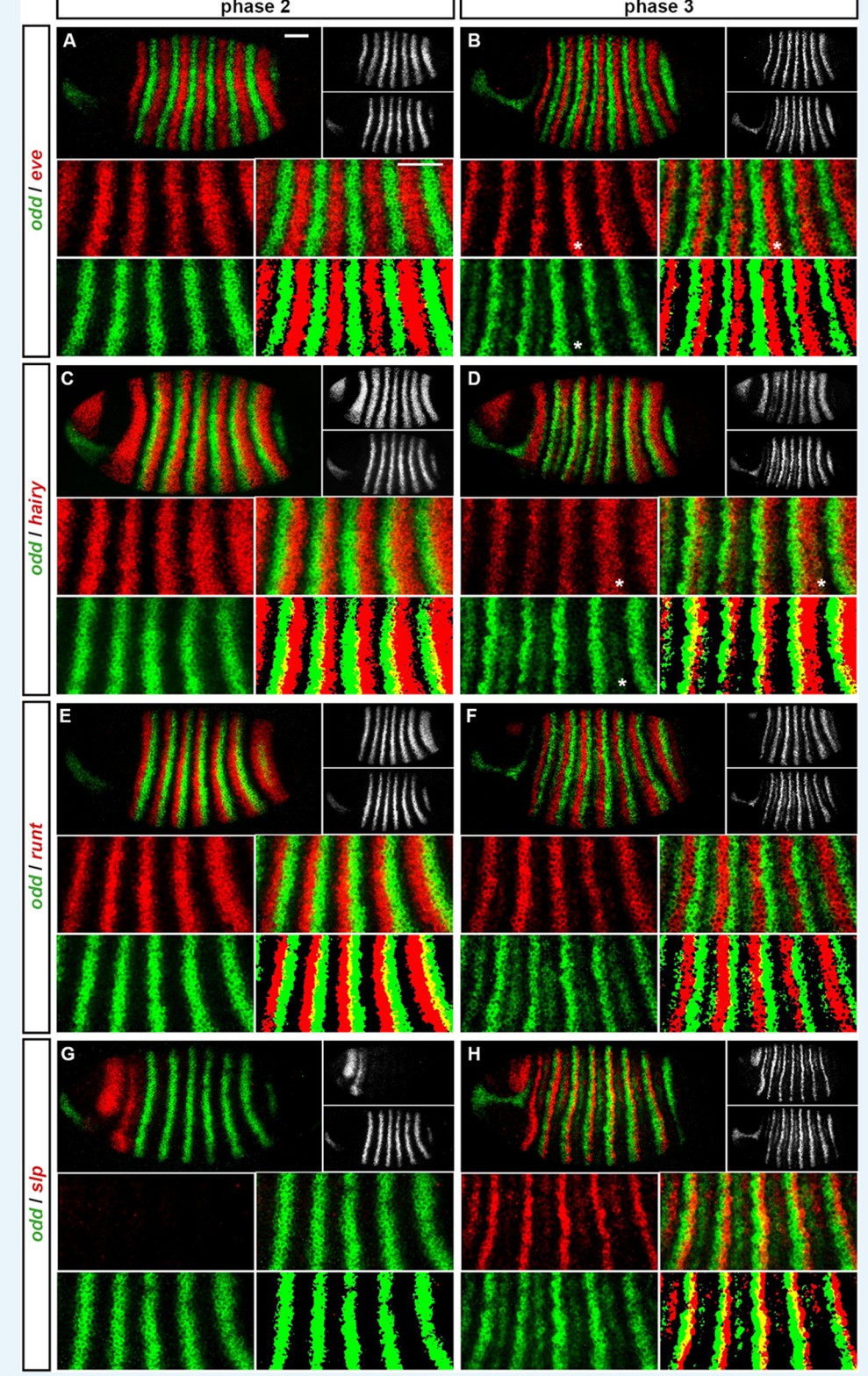

**Appendix 1—figure 1.** Expression of *odd* at phase 2 *versus* phase 3. Relative expression of *odd* and other pair-rule genes (**A**, **B** – *eve*; **C**, **D** – *hairy*; **E**, **F** – *runt*; **G**, **H** - *slp*) is shown in late phase 2 embryos (**A**, **C**, **E**, **G**) and in early phase 3 embryos (**B**, **D**, **F**, **H**). Individual channels are shown to the right of each whole embryo double channel image (*odd* bottom, other gene top). Other panels show blow-ups of expression in stripes 2–6 (individual channels, double channel image, and thresholded double channel image). *odd* expression is always shown in

green. *odd* expression overlaps with *eve* and *hairy* at phase 3 (e.g. asterisks marking nascent secondary stripe expression in **B**, **D**) but not at phase 2 (**A**, **C**). *odd* expression overlaps with *runt* at phase 2 (**E**) but not phase 3 (**F**). *slp* expression is absent for most of phase 2 (**G**) and is responsible for posterior narrowing of odd primary stripes at phase 3 (**H**). Scale bars = 50 μm. See text for details.

The following figure supplement is available for figure 11:
Appendix 1—Figure 1 supplement 1. Model for the regulation of *odd* transcription at phase 2 *versus* phase 3.

Thus there appear to be multiple changes to the regulation of *odd* between phase 2 and phase 3 (*Appendix 1—figure 1—figure supplement 1*): loss of repression by Eve and Hairy, and gain of repression by Runt, and possibly Prd. The lack of repression by Eve and Hairy does not compromise the late patterning of the primary *odd* stripes, because their patterning roles are taken over by new repressors. Slp protein appears at the end of cellularisation and takes over from Hairy at the posterior boundaries (*Appendix 1—figure 1H*; *Jaynes and Fujioka, 2004*). The new repression from Runt (and later, from En) seems to take over from Eve at the anterior boundaries (Appendix 2).

## sloppy-paired (*Appendix 1—figure 2*; *Appendix 1—figure 2—figure supplement 1*)

The primary stripes of *slp* appear at the end of phase 2, while the secondary stripes appear shortly afterwards, at the beginning of phase 3. In contrast to the other pair-rule genes, *slp* stripes are static and stable, with dynamic pattern refinements restricted to the head region. The *slp* locus has a large, complex regulatory region, with many partially redundant enhancer elements (*Fujioka and Jaynes, 2012*). A detailed study of two of these elements showed that the primary stripes are mediated by one element, while the secondary stripes require an additional enhancer that interacts non-additively with the first element (*Prazak et al., 2010*).

The primary stripes of *slp* are thought to be patterned by repression from Eve at their posteriors and repression by the combination of Runt and Ftz at their anteriors (*Swantek and Gergen, 2004*). There is plentiful evidence for repression of *slp* by Eve throughout segmentation (*Appendix 1—figure 2A,B*; *Fujioka et al., 1995*; *Riechmann et al., 1997*; *Jaynes and Fujioka, 2004*; *Swantek and Gergen, 2004*; *Prazak et al., 2010*). However, while the posterior boundaries of the Runt primary stripes do appear to define the anterior boundaries of the *slp* primary stripes (*Appendix 1—figure 2C*), we are not convinced that Runt and Ftz act combinatorially to repress *slp* (*Appendix 1—figure 2—figure supplement 2*).

We find that in *ftz* mutant embryos, the *slp* primary stripes form fairly normally during phase 2, with their anterior boundaries still seemingly defined by Runt, rather than expanding anteriorly to overlap the (Eve-negative) posterior halves of the *runt* stripes. Ectopic *slp* expression does not appear until phase 3. This indicates that Runt is able to repress *slp* in the absence of Ftz, at least temporarily. We therefore propose that during phase 2, *slp* is repressed by both Eve and Runt, regardless of whether Ftz is present, and that the anterior boundaries of the *slp* primary stripes are initially patterned by Runt alone.

In wild-type embryos, the *slp* secondary stripes appear at phase 3, in the anterior halves of the *runt* stripes (*Appendix 1—figure 2D*). There are competing models for how they are regulated. One model proposes that they are activated by Runt, but repressed by the combination of Runt and Ftz, so that their anterior boundary is defined by Runt and their posterior boundary is defined by Ftz (*Swantek and Gergen, 2004*; *Prazak et al., 2010*). A different model proposes that their anterior boundaries are defined by repression by Eve, while their posterior boundaries are defined by repression by Odd (*Jaynes and Fujioka, 2004*).

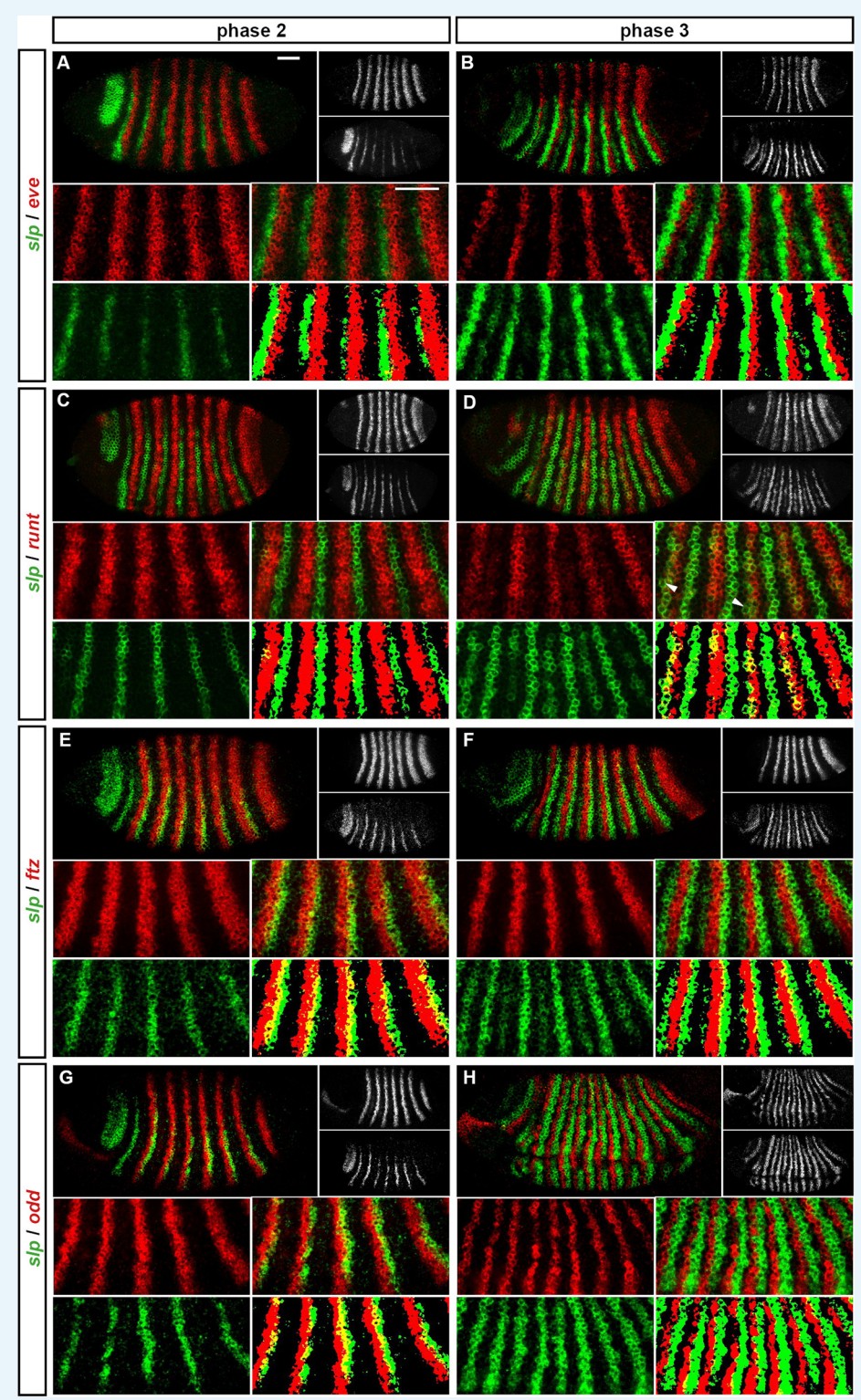

**Appendix 1—figure 2.** Expression of *slp* at phase 2 versus phase 3. Relative expression of *slp* and other pair-rule genes (**A**, **B** – *eve*; **C**, **D** – *runt*; **E**, **F** – *ftz*; **G**, **H** - *odd*) is shown in late phase 2 embryos (**A**, **C**, **E**, **G**) and in early phase 3 embryos (**B**, **D**, **F**, **H**). Individual channels are shown to the right of each whole embryo double channel image (*slp* bottom, other gene top). Other panels show blow-ups of expression in stripes 2–6 (individual channels, double channel image, and thresholded double channel image). *slp* expression is always shown in green. *slp*

expression overlaps with *runt* at phase 3 (**D**) but not at phase 2 (**C**). *slp* expression overlaps with *ftz* and *odd* at phase 2 (**E**, **G**) but not phase 3 (**F**, **H**). *slp* expression never overlaps with *eve* (**A**, **B**). Arrowheads in (**D**) indicate cells where *slp* secondary stripe expression does not coincide with *runt* expression. Scale bars = 50 μm. See text for details.

The following figure supplement is available for figure 12:
Appendix 1—Figure 2 supplement 1. Model for the regulation of s*lp* transcription at phase 2 *versus* phase 3.

Appendix 1—Figure 2 supplement 2. Runt represses *slp* during phase 2 in both wild-type and *ftz* mutant embryos.

The posterior borders of the *eve* primary stripes abut the anterior borders of the *runt* primary stripes during early phase 3 (***Appendix 1—figure 3F***). Mutual repression between Eve and Runt (***Gergen and Butler, 1988***; ***Manoukian and Krause, 1992***; ***Manoukian and Krause, 1993***; ***Klingler and Gergen, 1993***) temporarily stabilises these expression boundaries, which also correspond to the anterior boundaries of the *slp* secondary stripes. Because of the regulatory feedback between Eve and Runt, the distinct regulatory hypotheses of repression by Eve *versus* activation by Runt actually predict identical effects on the expression of *slp* in a variety of genetic backgrounds. Therefore, much of the experimental evidence cited in favour of each of these models does not really discriminate between them.

When we look carefully at the early expression of the *slp* secondary stripes, we occasionally find *slp* expression in a *runt*-negative cell (arrowheads in ***Appendix 1—figure 2D***), but we never observe cells expressing both *eve* and *slp* (***Appendix 1—figure 2B***, and data not shown). This indicates that Eve directly patterns the anterior boundaries of the *slp* secondary stripes, while the regulatory role of Runt is indirect. Consistent with this hypothesis, a reporter study found that Runt did not appear to directly regulate a *slp* enhancer that drives 14 stripes at phase 3 (***Sen et al., 2010***; ***Fujioka and Jaynes, 2012***).

While *ftz* and *odd* are subject to similar regulation during phase 2 and consequently have similar expression domains, the slightly broader Ftz stripes appear to define the posterior boundary of *slp* secondary stripe expression (***Appendix 1—figure 2F***). This does not rule out Odd as a repressor of *slp*, however. Indeed, experimental evidence supports direct repression of *slp* by Odd (***Saulier-Le Dréan et al., 1998***) as well as by Ftz (***Nasiadka and Krause, 1999***; ***Swantek and Gergen, 2004***; ***Prazak et al., 2010***). Repression from Odd is likely to stabilise the anterior boundaries of both sets of *slp* stripes during late phase 3 (***Appendix 1—figure 2H***).

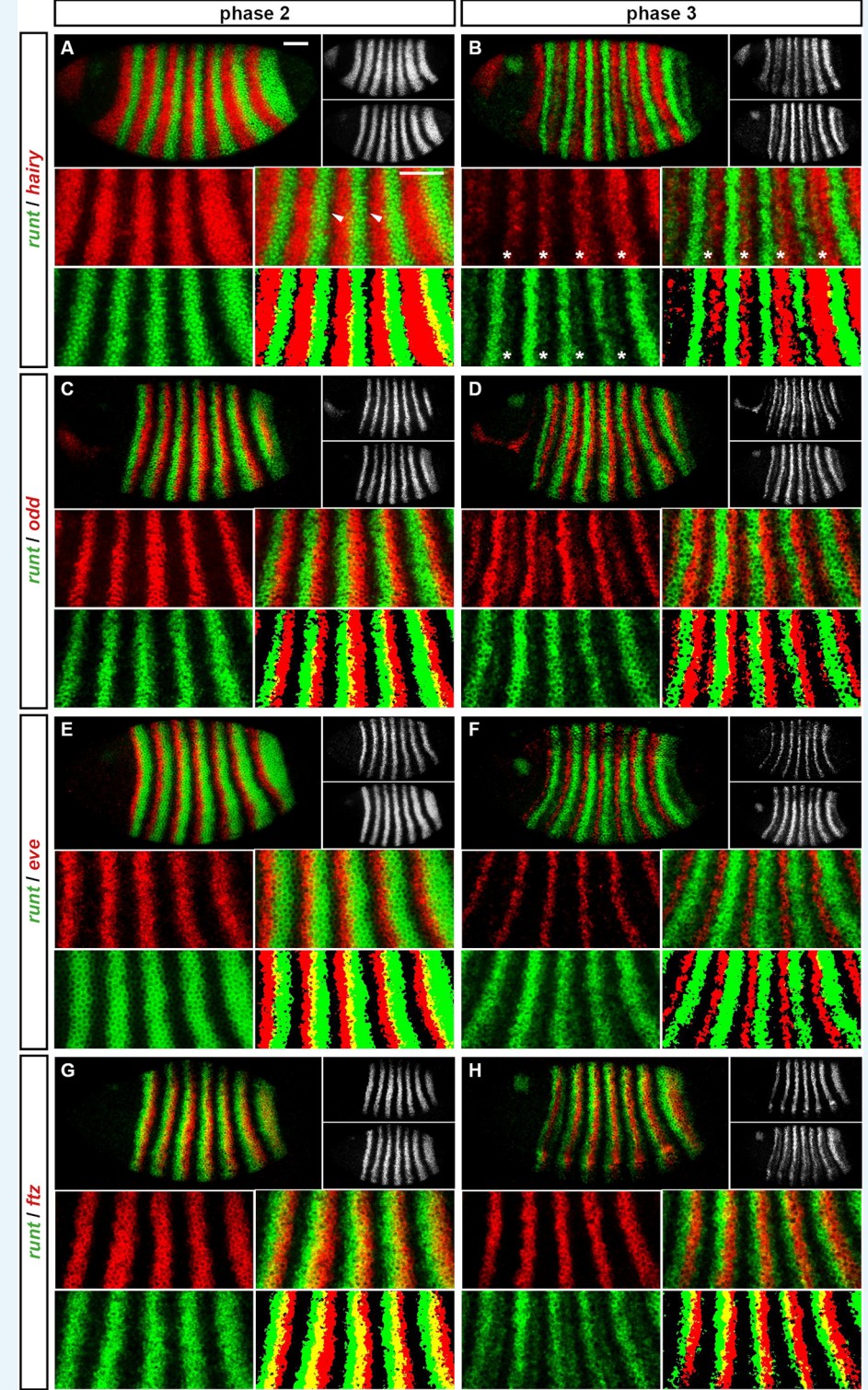

**Appendix 1—figure 3.** Expression of *runt* at phase 2 *versus* phase 3. Relative expression of *runt* and other pair-rule genes (**A**, **B** – *hairy*; **C**, **D** – *odd*; **E**, **F** – *eve*; **G**, **H** - *ftz*) is shown in late phase 2 embryos (**A**, **C**, **E**, **G**) and in early phase 3 embryos (**B**, **D**, **F**, **H**). Individual channels are shown to the right of each whole embryo double channel image (*runt* bottom, other gene top). Other panels show blow-ups of expression in stripes 2–6 (individual channels, double channel image, and thresholded double channel image). *runt* expression is always shown in green. *runt* primary stripes are out of phase with *hairy* (**A**) but *runt* secondary stripes (asterisks in **B**) emerge within domains of *hairy* expression. *runt* expression overlaps with *odd* and *eve* at

phase 2 (**C**, **E**) but not phase 3 (**D**, **F**). *runt* expression overlaps with *ftz* at both phase 2 and phase 3 (**G**, **H**). Arrowheads in (**A**) point to clear gaps between the posterior boundaries of the *runt* stripes and the anterior boundaries of the *hairy* stripes. Scale bars = 50 µm. See text for details.

The following figure supplement is available for figure 13:
Appendix 1—Figure 3 supplement 1. Model for the regulation of *runt* transcription at phase 2 *versus* phase 3.

We see no compelling evidence that the repressive activity of Ftz on *slp* is mediated by Runt. It is clear that the presence or absence of Runt has dramatic effects on the expression pattern of *slp*, and that this is modified by the presence or absence of Ftz (***Swantek and Gergen, 2004***; ***Prazak et al., 2010***). However, we think that these effects are likely to be explained either by indirect interactions or by the repressive role of Runt during phase 2 (see above).

We thus conclude that regulation of *slp* undergoes several changes at phase 3 (***Appendix 1— figure 2—figure supplement 1***). Repression by Runt is lost, while repression by Ftz and Odd is gained. We find no evidence for direct activation of *slp* by Runt, nor do we find evidence for a combinatorial interaction between Ftz and Runt (*cf.* ***Swantek and Gergen, 2004***). Instead, we think that their roles are temporally separate, with Runt acting at phase 2 and Ftz acting at phase 3.

## *runt* (*Appendix 1—figure 3*; *Appendix 1—figure 3—figure supplement 1*)

During phase 2, the primary stripes of *runt* are broadly out of phase with those of *hairy* (***Appendix 1—figure 3A***). There is good evidence for repression of *runt* by Hairy (***Gergen and Butler, 1988***; ***Klingler and Gergen, 1993***; ***Jiménez et al., 1996***), and it is commonly thought that Hairy defines both the anterior and posterior boundaries of *runt* expression (e.g. ***Edgar et al., 1989***; ***Schroeder et al., 2011***). However, we find clear gaps between the posterior boundaries of *runt* expression and the anterior boundaries of *hairy* expression (arrowheads in ***Appendix 1—figure 3A***), indicating that some other pair-rule gene must be repressing *runt* from the posterior. We propose that the posterior boundaries of the *runt* primary stripes are defined by repression from Odd (***Appendix 1—figure 3C***). This hypothesis is strongly supported by the observations that the *runt* stripes widen slightly in *odd* mutant embryos and are directly repressed by ectopic Odd (***Saulier-Le Dréan et al., 1998***).

During phase 3, new *runt* expression appears to the posterior of the primary stripes, and gradually intensifies to form the secondary stripes. At the same time, the primary stripes narrow from the posterior, producing a 'splitting' of the broadened *runt* domains (***Klingler and Gergen, 1993***). The two sets of stripes are initially driven by different enhancers, although each of the two enhancers later drive 14 segmental stripes during germband extension (***Klingler et al., 1996***). This indicates that the primary and secondary *runt* stripes are subject to different regulatory logic during phase 3.

During cellularisation, the anterior of each *runt* stripe overlaps with *eve* expression (***Appendix 1—figure 3E***), and accordingly Eve does not appear to repress *runt* during this stage (***Manoukian and Krause, 1992***). However, Eve starts to repress *runt* at phase 3 (***Manoukian and Krause, 1992***; ***Klingler and Gergen, 1993***). Eve appears to act on both sets of *runt* stripes, defining the posterior boundaries of the secondary stripes as well as the anterior boundaries of the primary stripes (***Appendix 1—figure 3F***).

It has been hypothesised that the narrowing of the *runt* primary stripes is caused by direct repression by Ftz (***Klingler and Gergen, 1993***; ***Wolff et al., 1999***). However, this is not supported by Ftz misexpression (***Nasiadka and Krause, 1999***). Indeed, we find that the posteriors of the *runt* primary stripes continue to overlap with the anteriors of the *ftz* stripes

for a considerable period during phase 3, ruling out direct repression by Ftz (*Appendix 1—figure 3H*). Instead, the posteriors of the *runt* primary stripes appear to be repressed by the even-numbered En stripes, which are activated by Ftz (*Klingler and Gergen, 1993*; *DiNardo and O'Farrell, 1987*). Before the appearance of En protein, the posterior boundaries continue to be defined by repression from Odd (*Appendix 1—figure 3D*).

We have not investigated whether Hairy continues to repress the regulatory element driving the *runt* primary stripes during phase 3, although it is possible it does not. However, it is clear that Hairy does not repress the element driving the *runt* secondary stripes, because they are located within domains of *hairy* expression (*Appendix 1—figure 3B*). The secondary stripes also overlap with Odd expression (*Appendix 1—figure 3D*), indicating that, unlike the primary stripes, they are not sensitive to repression by Odd.

It is not clear what defines the anterior boundaries of the *runt* secondary stripes. The locations of these stripes correlate very closely with those of the *slp* primary stripes, in both wild-type and *ftz* mutant embryos (see *Appendix 1—figure 2—figure supplement 2*). However, because *runt* expression is not noticeably affected in *slp* mutant embryos (*Klingler and Gergen, 1993*), this must result from shared regulation rather than a patterning role for Slp itself. Indeed, Eve defines the posterior boundaries of both the *slp* primary stripes and the *runt* secondary stripes (see above). The anterior boundaries of the *slp* primary stripes are defined by repression by the Runt primary stripes (see above), raising the possibility that the *runt* secondary stripes are regulated in the same way, at least initially. If true, this would be the first example of direct autorepression by a pair-rule gene during segmentation.

Finally, Prd is required for the expression of the secondary stripes (*Klingler and Gergen, 1993*). Prd appears to provide general activatory input to the element driving the stripes, but is unlikely to convey specific positional information, because the expression boundaries of the Prd stripes do not correspond to those of the *runt* secondary stripes (*Figure 6B*). Prd is also unlikely to provide temporal information to the element: the expression of the *runt* secondary stripes is delayed relative to the appearance of Prd protein (*Pisarev et al., 2009*), suggesting that Prd alone is not sufficient for their activation.

In summary, there is one important change to the regulation of the *runt* zebra element at phase 3 (*Appendix 1—figure 3—figure supplement 1*). Repression by Eve is gained, and may potentially replace repression by Hairy. In addition, a separate element driving the secondary stripes begins to be expressed at phase 3. This element appears to be repressed by Eve and perhaps Runt, and activated by Prd.

### even-skipped

*eve* does not possess a zebra element active during phase 2, and therefore its regulation does not come under control of the pair-rule network until its 'late' element turns on at phase 3. This element generates strong expression in the anterior halves of the pre-existing early *eve* stripes. The posterior boundaries of the late stripes are temporarily defined by repression by Runt, while the anterior boundaries are defined by repression by Slp (*Appendix 1—figure 2B*; *Appendix 1—figure 3F*; *Jaynes and Fujioka, 2004*). Odd also represses late *eve* (*Saulier-Le Dréan et al., 1998*), and will temporarily compensate for the lack of repression by Slp in *slp* mutant embryos (*Jaynes and Fujioka, 2004*). The late *eve* stripes do not persist long after gastrulation, largely owing to the appearance of En protein, another repressor of *eve* (*Harding et al., 1986*).

In addition to the strong 'major' stripes at the anteriors of the odd-numbered parasegments, faint 'minor' stripes of *eve* expression appear during gastrulation in the anteriors of the even-numbered parasegments (*Macdonald et al., 1986*; *Frasch et al., 1987*; *Figure 6C*). These stripes are also driven by the late element (*Fujioka et al., 1995*), and are therefore likely to share the same regulatory logic as the major stripes. They do not appear to play any role in patterning, since deletions of the *eve* late element do not affect the patterning of the even-numbered parasegment boundaries (*Fujioka et al., 1995*; *Fujioka et al., 2002*).

## Other pair-rule genes

In contrast to the other pair-rule genes, *hairy* and *ftz* do not show signs of significantly altered spatial regulation at gastrulation (*Figure 6*). The *hairy* stripes, which are regulated by stripe-specific elements, begin to fade away. During phase 2, the anterior boundaries of the *ftz* stripes are defined by repression by Eve, while the posterior boundaries are defined by repression by Hairy (*Ish-Horowicz and Pinchin, 1987*; *Carroll et al., 1988*; *Frasch et al., 1988*; *Gergen and Butler, 1988*; *Vavra and Carroll, 1989*; *Manoukian and Krause, 1992*; *Jiménez et al., 1996*). The *ftz* stripes narrow from the posterior at phase 3, but this appears to be simply due to the new appearance of Slp protein, which also represses *ftz* (*Cadigan et al., 1994b*), rather than evidence for altered regulatory logic (*Figure 6B*). Autoregulation is likely to play a role in maintaining the late *ftz* expression pattern (*Hiromi and Gehring, 1987*; *Schier and Gehring, 1992*), perhaps indicating that sustained repression of *ftz* expression within the interstripes by other pair-rule proteins may not be strictly necessary.

## Appendix 2

# The patterning of the anterior borders of the *odd* primary stripes

### Aetiology of the ftz/odd expression offsets

One particularly intriguing feature of *opa* mutant embryos is that the offset between the anterior boundaries of the *ftz* and *odd* stripes is largely absent (**Benedyk et al., 1994**; *Appendix 2—figure 1*). In wild-type embryos, the anterior boundaries of the *odd* primary stripes are shifted posteriorly relative to those of the *ftz* stripes by about one cell row. This relative phasing is important for patterning the even-numbered *en* stripes, which are activated by Ftz but repressed by Odd (**Coulter et al., 1990**; **Manoukian and Krause, 1992**; **Mullen and DiNardo, 1995**).

The offsets between the anterior boundaries of *ftz* and *odd* require the presence of the early Eve stripes (**Fujioka et al., 1995**). It is thought that the posterior halves of these stripes act as morphogen gradients that repress *odd* at lower concentrations of Eve than required to repress *ftz*, and thus differentially position the expression domains of the two genes (**Fujioka et al., 1995**; **Manoukian and Krause, 1992**). We find this explanation unsatisfactory, for two reasons.

First, a careful analysis of wild-type gene expression calls into question the hypothesis that the early Eve stripes are functioning in this manner. Both *ftz* and *odd* lack a stripe-specific element for stripe 4, and so the expression seen in these stripes is a true reflection of regulatory control by pair-rule proteins, whereas inferences from the remaining stripes are complicated by gap protein-regulated contributions to the overall expression pattern. When the zebra element-driven expression of *ftz* and *odd* kicks in and stripe 4 appears, clear one cell wide offsets are seen at the anterior borders of most of the stripes, but are absent from stripe 4 (*Appendix 2—figure 1A*). This suggests that Eve is not differentially regulating the two genes, and that the offsets that are seen in the other stripes are instead generated by bespoke positioning of individual stripes by stripe-specific elements.

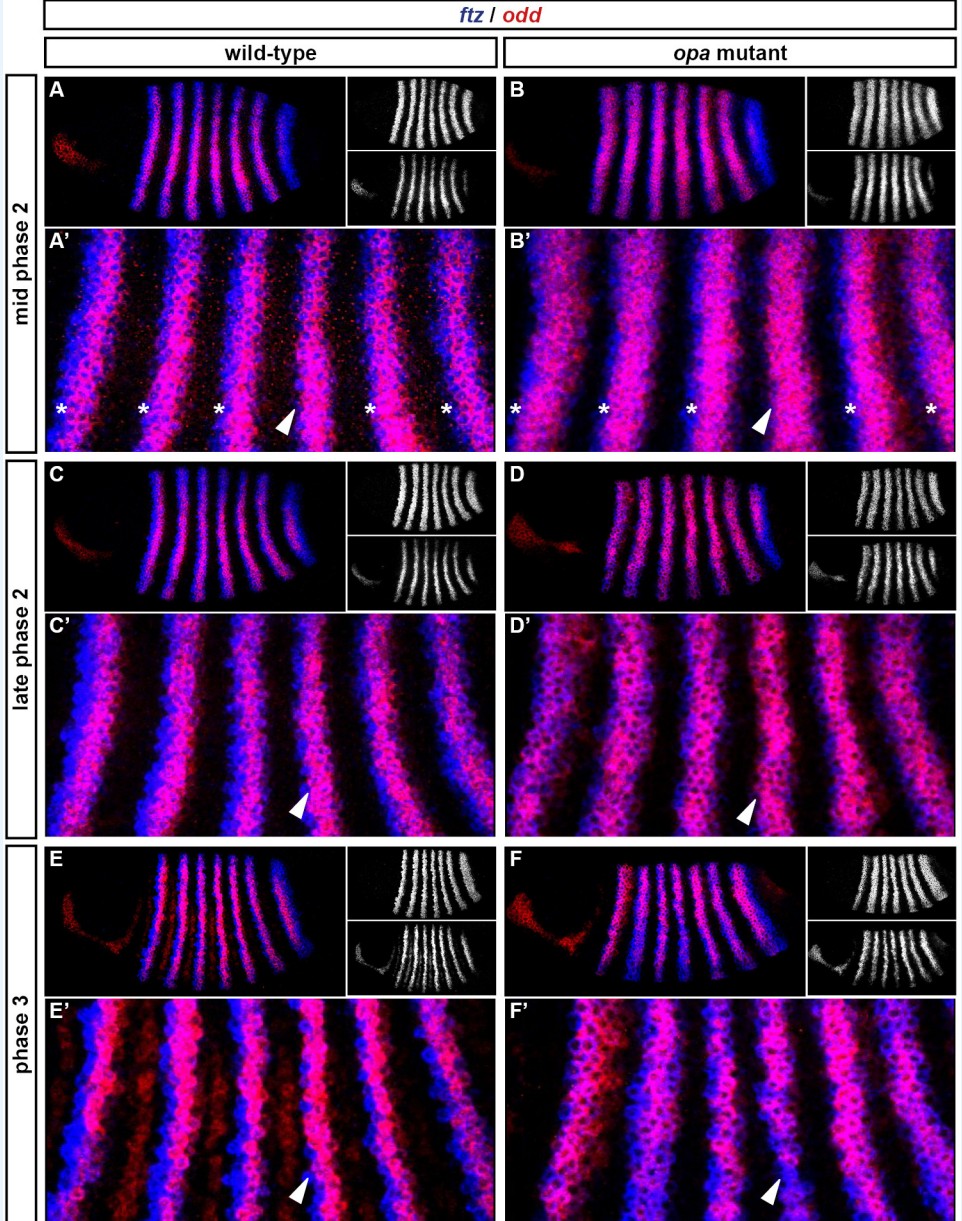

**Appendix 2—figure 1.** The *ftz/odd* anterior boundary offsets are lost in *opa* mutant embryos at gastrulation. Relative expression of *ftz* and *odd* in wild-type and *opa* mutant embryos. (**A–D**) Whole embryos, anterior left; (**A–C**) show lateral views, (**D**) shows a ventral view. Single channels are shown to the right of each double channel image (*ftz* top, *odd* bottom). (**A'–D'**) Blow-ups of stripes 1–6. Arrowheads point to stripe 4, for which neither *ftz* nor *odd* possesses a stripe-specific element. Asterisks in (**A'**, **B'**) indicate early *ftz/odd* offsets in stripes where *ftz* expression is partially driven by stripe-specific elements. Scale bars = 50 µm.

The following figure supplements are available for figure 14:

Appendix 2—Figure 1 supplement 1. The *odd* primary stripes expand anteriorly in *opa* mutant embryos.

Appendix 2—Figure 1 supplement 2. Model for the patterning of the anterior boundaries of *ftz* and *odd*.

Second, maintenance of the offsets between *ftz* and *odd* expression seems to require Opa function. In wild-type embryos, offsets are observed from late cellularisation onwards for all

stripes, including stripe 4 (*Appendix 2—figure 1C,E*), indicating that *ftz* and *odd* must be differentially regulated by pair-rule proteins during these later stages. In *opa* mutant embryos, we find that the relative phasing of *ftz* and *odd* appears normal at mid-cellularisation, with offsets present for most stripes, but absent for stripe 4 (*Appendix 2—figure 1B*), as in wild-type. By late-cellularisation, however, the anterior boundaries of the two sets of stripes tend to coincide (*Appendix 2—figure 1D*). We therefore do not think that the early Eve stripes can be directly patterning the offsets, because early *eve* expression is normal in *opa* mutant embryos. Late *eve* expression is lost in *opa* mutant embryos (see above), but this phase of expression cannot be regulating the pattern either, because *eve* rescue constructs lacking the *eve* late element still produce the offsets (*Fujioka et al., 1995*). Therefore, the maintenance of offsets in stripes 1–3 and 5–7, and the establishment of the offset in stripe 4, must be patterned by a pair-rule protein other than Eve, by way of an Opa-dependent regulatory interaction.

Coincident anterior boundaries of *ftz* and *odd* could be produced by a posterior retraction of *ftz* expression, or alternatively by an anterior expansion of *odd* expression. We interpret the patterns in *opa* mutant embryos as representing the latter scenario. The *odd* stripes still share posterior boundaries with the *ftz* stripes, but appear wider than in wild-type embryos, consistent with de-repression at the anterior (*Appendix 2—figure 1C,D*). Furthermore, when we compare phasings of the *odd* stripes with those of *eve*, the domains of *odd* expression appear significantly anteriorly expanded in *opa* mutant embryos compared to wild-type (*Appendix 2—figure 1—figure supplement 1*)

Following from this reasoning, it appears that the *ftz/odd* offsets observed at late cellularisation in wild-type embryos must be caused by anterior repression of *odd* (and not *ftz*) by an appropriately-located pair-rule protein in combination with Opa. We suggest that this protein is Runt. Above, we hypothesised that in wild-type embryos, Runt starts to repress *odd* at phase 3 (or more accurately, given the expression data in *Appendix 2—figure 1*, at late phase 2), thus defining the anterior boundaries of the *odd* primary stripes (*Appendix 1—figure 1*; *Appendix 1—figure 1—figure supplement 1*). We also identified Opa as being required for the regulatory changes observed at phase 3 (*Figure 8*; *Figure 8—figure supplement 4*).

This new model (*Appendix 2—figure 1—figure supplement 2*) explains the observations from *opa* mutants. In the absence of Opa activity, Runt fails to repress *odd*, and the anterior boundaries of *odd* expression presumably continue to be defined by the posterior boundaries of the Eve stripes, which also define the anterior boundaries of the *ftz* stripes. This results in the loss of the *ftz/odd* offsets that pattern even-numbered *en* stripes in wild-type.

## Opa spatially patterns odd stripe 7

We noticed that in *opa* mutant embryos, *odd* stripe 7 appears to expand both anteriorly and ventrally (*Appendix 2—figure 2C,H*). *odd* stripe 7 is both spatially and temporally unusual: it is not expressed dorsally or ventrally, and it first appears considerably after the other six *odd* stripes have been established. In fact, it is the only primary pair-rule stripe to appear after the trunk stripes of the secondary pair-rule gene *prd* are established (*Appendix 2—figure 2—figure supplement 1*).

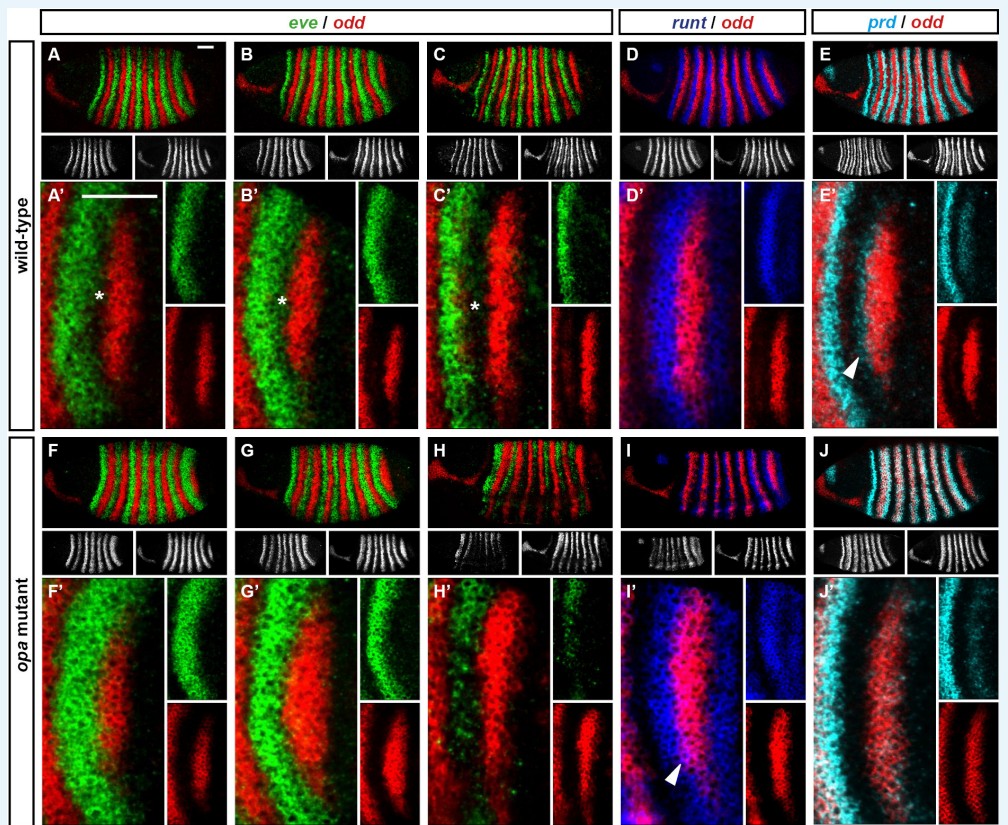

**Appendix 2—figure 2.** *odd* stripe 7 expands anteriorly and ventrally in *opa* mutants. Expression of *odd* relative to that of *eve*, *runt* and *prd*, in wild-type and *opa* mutant embryos. (**A–J**) Whole embryos, individual channels shown below each double channel image (*odd* right). (**A**, **F**) Late phase 2; (**B–E**, **G–J**) early phase 3. (**A'–J'**) Blow-ups of stripe 7 region (images rotated so that stripes appear vertical). (**A'–C'**, **F'–H'**) The anterior boundary of *odd* stripe 7 remains correlated with the posterior boundary of *eve* stripe 7 during phase 3 in *opa* mutant embryos, but not in wild-type. Asterisks in (**A–C**) indicate regions free of both *eve* and *odd* expression. Note that in *opa* mutant embryos, the *eve* stripes gradually fade away, while in wild-type they narrow from the posterior but remain strongly expressed. (**D'**, **I'**) *odd* stripe 7 expands anteriorly relative to *runt* stripe 7 in *opa* mutant embryos. In wild-type embryos, *odd* expression does not overlap with *runt* expression after the posterior half of *runt* stripe 7 becomes repressed (**D'**). In *opa* mutant embryos, the anterior border of *odd* stripe 7 overlaps with *runt* expression (purple regions in **I'**). Arrowhead points to a conspicuous region of *odd*/*runt* co-expression. (**E'**, **J'**) *odd* stripe 7 expands anteriorly relative to *prd* expression in *opa* mutant embryos. Arrowhead in (**E'**) points to *prd* expression anterior to *odd* stripe 7. Scale bars = 50 μm.

The following figure supplements are available for figure 15:
**Appendix 2—Figure 2 supplement 1.** *odd* stripe 7 appears after the primary stripes of *prd*, but before the primary stripes of *slp*.

**Appendix 2—Figure 2 supplement 2.** The posterior border of *eve* stripe 7 shifts anteriorly relative to the anterior border of *odd* stripe 7.

**Appendix 2—Figure 2 supplement 3.** Model for the patterning of the anterior boundaries of the *odd* primary stripes.

We have described above how the anterior boundaries of the *odd* stripes are defined first by repression by Eve, and subsequently by repression by Runt, which requires the presence of Opa (*Appendix 2—figure 1—figure supplement 2*). When *odd* stripe 7 first appears, its anterior boundary correlates well with the posterior boundary of *eve* expression, and is likely be patterned by repression by Eve (*Appendix 2—figure 2—figure supplement 2C*). The posterior boundary of *eve* stripe 7 then markedly shifts anteriorly, while *odd* stripe 7 remains static, suggesting that its anterior boundary is maintained by repression from some other protein (*Appendix 2—figure 2—figure supplement 2D*). However, the seventh stripe of *runt* is abnormally broad and completely encompasses the domain of *odd* expression (*Appendix 2—figure 2—figure supplement 2B,D*). Consequently, Runt cannot be providing spatial information to *odd* in this region of the embryo. It is therefore not clear which protein spatially delimits the anterior boundary of *odd* stripe 7 at gastrulation.

We suggest that it is actually Opa that patterns the anterior boundary of *odd* stripe 7. *odd* is repressed by the combination of Runt and Opa, but not by either gene alone. Theoretically, it makes no difference which protein provides the spatial information to pattern an expression domain of *odd,* as long as the repressive activity of the co-expressed proteins is appropriately localised. For *odd* stripes 2–6, Opa is expressed ubiquitously, while Runt is patterned. For *odd* stripe 7, we find that the position of its anterior boundary is prefigured by the posterior boundary of the broad *opa* expression domain (*Figure 7B–E*). Therefore, in the posterior of the embryo the situation seems to be the other way around: Runt is expressed ubiquitously, while Opa provides the necessary spatial information (*Appendix 2—figure 2—figure supplement 3*).

Because *odd* stripe 7 is so delayed relative to the other primary pair-rule stripes, there is only a short time between its appearance and the first signs of Opa regulatory activity in the embryo. Therefore, while the early expression of *odd* stripe 7 is likely to be patterned by Eve, repression by Runt + Opa would soon take over, explaining why *odd* stripe 7 remains static rather than shifting anteriorly in concert with *eve*. Accordingly, we observe that in *opa* mutant embryos, where the *odd* anterior boundaries are presumably defined by Eve at all times, *odd* stripe 7 expands both anteriorly and ventrally over time, correlating well with the shifting posterior boundary of *eve* stripe 7 (*Appendix 2—figure 2*). Indeed, in *opa* mutant embryos the anterior boundary of *odd* 7 is located at a similar position to the anterior boundary of *prd* stripe 8 (also likely to be defined by repression by Eve), whereas in wild-type it is offset from it posteriorly (*Appendix 2—figure 2E,J*).

The distinctive shape of *odd* stripe 7 can therefore be explained by the curvature of the *opa* posterior boundary. Thus, in the posterior of the embryo, Opa conveys both temporal and spatial information to the segmentation process.

