## [Decision Letter]

Thank you for submitting your article "Odd-paired controls frequency doubling in *Drosophila* segmentation by altering the pair-rule gene regulatory network" for consideration by *eLife*. Your article has been reviewed by three peer reviewers, one of whom is a member of our Reviewing Editors, and the evaluation has been overseen by a Reviewing Editor and K VijayRaghavan as the Senior Editor. One of the three reviewers has agreed to reveal his identity: Siegfried Roth (Reviewer #3).

The reviewers have discussed the reviews with one another and the Reviewing Editor has drafted this decision to help you prepare a revised submission.

Summary:

This paper addresses one of the major unresolved questions of the *Drosophila* segmentation cascade: the transition form pair-rule to segment polarity gene expression, which is accompanied by frequency-doubling creating the final segment number and the polarity within each segment. For long time it has been recognized that compared to earlier steps of the segmentation cascade this last step is particularly complex and in essential ways not understood. The paper by Clark and Akam has undertaken a heroic effort by re-addressing this fundamental question. The authors first characterize the transition through detailed description of the dynamic changes in gene expression at the relevant stages (late blastoderm to early germ band extension). The phenomenological part of the paper suggests a coordinated stage-specific rewiring of the pair-rule gene network which cannot be explained solely on the basis hierarchical feed-forward actions of the pair-rule genes, but rather implies an independent element of temporal control. This element is provided by Odd-paired (Opa), the only pair-rule gene with non-striped largely uniform expression, as the authors demonstrate by analyzing the expression changes of striped pair-rule genes in an *opa* mutant background. Opa had been largely neglected in former studies labeling it as a permissive factor without instructive function. By showing that *opa* is required for a temporal switch in regulatory logic the authors place *opa* at the center stage and highlight the importance of temporal control for complex spatial patterning.

The reviewers agree that this work was very nicely done and a major contribution to the field. It provides a new concept that will be important for modelers who will want to describe fly patterning in its entirety and build on it as the paradigm of a dynamic gene network. The presentation of the work could be improved by a few revisions, however.

Essential revisions:

1) Move the discussion of individual genes in the first section of the Results to a supplemental section. Please also modify accordingly the paragraph currently preceding this section that says that it need not be understood to follow the rest of the paper.

2) Please double check references are correct historically. For example, Pankratz & Jackle 1990 are credited with early establishment of individual pair-rule stripes, but a reviewer pointed out that the Levine lab showed this earlier for eve.

3) Please clarify the following: Opa has an effect on the splitting of some of the pair-rule genes, although its pair-rule phenotype cannot be explained by its function as a timer of stripe splitting. The authors suggest that this pair-rule effect exist because *opa* affects the 'autoregulatory' element of eve, which cannot be activated in the mutant. However, it is not clear how *opa* controls this element. Also, is the *opa* pair-rule phenotype similar to the loss of the autoregulatory element?

4) Please reduce some of the redundancy within the Discussion section and between the Discussion and Results sections.

---

## [Author Response]

*1) Move the discussion of individual genes in the first section of the Results to a supplemental section. Please also modify accordingly the paragraph currently preceding this section that says that it need not be understood to follow the rest of the paper.*

We have moved this section to an appendix, retaining only the discussion of Odd/*prd* interaction in the main text as an illustrative example. We have also moved the final sections of the results (which discussed detailed aspects of the regulation of the *odd* primary stripes) to a second appendix, as they are tangential to the main thrust of the paper and built on material established in the first appendix.

*2) Please double check references are correct historically. For example, Pankratz & Jackle 1990 are credited with early establishment of individual pair-rule stripes, but a reviewer pointed out that the Levine lab showed this earlier for eve.*

The paper we cited was a review of the early literature on this topic; we have added references to the primary research papers. We have also added extra references on the segment-polarity network.

*3) Please clarify the following: Opa has an effect on the splitting of some of the pair-rule genes, although its pair-rule phenotype cannot be explained by its function as a timer of stripe splitting. The authors suggest that this pair-rule effect exist because opa affects the 'autoregulatory' element of eve, which cannot be activated in the mutant. However, it is not clear how opa controls this element. Also, is the opa pair-rule phenotype similar to the loss of the autoregulatory element?*

We were not sufficiently clear regarding the aetiology of the *opa* pair-rule phenotype and have added a section at the end of the Results to address this question specifically. We have also expanded upon the functional roles of the late stripes in the Discussion.

The role of Opa in mediating stripe doubling does indeed explain the pair-rule phenotype, because the resulting segmental stripes perform segment-polarity functions during germband extension. However, the segmental patterns of the pair-rule genes are not involved in regulating the initial expression of the segment-polarity genes at the end of cellularisation, hence the confusion.

We did not intend to give the impression that the *opa* pair-rule phenotype is mediated by its effects on the *eve* late element. The *eve* late element plays a role in stabilising the odd-numbered parasegment boundaries, preventing *slp* and *wg* expression from invading the odd-numbered *en* stripes. Flies lacking the *eve* late element may show narrowing or loss of odd-numbered *en* stripes, but a proportion of individuals survive to adulthood, indicating that late Eve expression is not strictly required for segmentation (Fujioka et al., 2002). It is certainly possible that loss of late Eve expression results in similar phenotypic effects in *opa* mutant embryos, but such effects would be subtle relative to the other expression changes seen in these embryos, and we have not looked for them specifically.

As to how Opa controls the *eve* late element, phenomenologically it seems that Opa provides general activation (perhaps cooperatively with Prd), while Slp, Runt, Odd and En are spatially patterned repressors that override this activation. Concrete mechanistic details will require study of the enhancer element itself.

*4) Please reduce some of the redundancy within the Discussion section and between the Discussion and Results sections.*

We have streamlined the Results and Discussion by merging sections that covered similar ground, and excising repetitive material. We have also removed the paragraphs from the Discussion that speculated about concentration-dependent Opa activity, and replaced these with a section in the Results that provides evidence for this from gene expression patterns in *opa* hypomorphs.